# Evaluating the Adversarial Robustness of CNNs Layer by Layer

**Yaowen Wang**                                                            *yxw5684@psu.edu*
*Department of Electrical Engineering*
*The Pennsylvania State University*

**Daniel Cullina**                                                          *cullina@psu.edu*
*Department of Electrical Engineering*
*The Pennsylvania State University*

**Reviewed on OpenReview:** *https://openreview.net/forum?id=2Gx9KzsaYB*

## Abstract

In order to measure the adversarial robustness of a feature extractor, Bhagoji et al. introduced a distance on example spaces measuring the minimum perturbation of a pair of examples to achieve identical feature extractor outputs. They related these distances to the best possible robust accuracy of any classifier using the feature extractor. By viewing initial layers of a neural network as a feature extractor, this provides a method of attributing adversarial vulnerability of the classifier as a whole to individual layers. However, this framework views any injective feature extractor as perfectly robust: any bad choices of feature representation can be undone by later layers. Thus the framework attributes all adversarial vulnerabilities to the layers that perform dimensionality reduction. Feature spaces at intermediate layers of convolutional neural networks are generally much larger than input spaces, so this methodology provides no information about the contributions of individual layers to the overall robustness of the network. We extend the framework to evaluate feature extractors with high-dimensional output spaces by composing them with a random linear projection to a lower dimensional space. This results in non-trivial information about the quality of the feature space representations for building an adversarial robust classifier.

## 1 Introduction

Over the last several years, a line of work has characterized the optimal adversarial loss achievable for a particular data distribution and adversarial constraint specification (Bhagoji et al., 2019; 2021; Pydi & Jog, 2020; Trillos et al., 2023; Dai et al., 2023). These works characterize the problem of minimizing adversarial loss over all possible classifier functions to some kind of optimal transport problem on the class conditional distributions. In order to set up this problem, one needs the distance structure of the data distribution with regard to the distance appearing in the adversarial constraint. These characterizations of optimal loss provide a lower bound for loss achievable using a particular architecture and a benchmark performance level that a trained classifier can be compared to. At a higher level, they provide an objective method to determine whether a given adversarial constraint is reasonable to use with a particular dataset.

More recently, this method has been extended to analyze the robustness of a feature extractor, or equivalently the initial layers of a feed-forward network (Bhagoji et al., 2023). Just as the minimum loss on a data distribution is a minimization over all possible classifiers, the robustness of a feature extractor is a minimization over all classifiers using that feature extractor. The robustness of a feature extractor can be computed via a characterization that uses the distance structure of the data distribution using a specially defined distance that takes into account the behavior of the feature extractor. This allows the adversarial loss of the model to be decomposed layer-by-layer: including a layer of the model in the feature extractor decreases the size of

the class of models using that feature extractor and thus increases the minimum loss. This increase can be considered the contribution of that layer to the overall adversarial loss of the model.

Unfortunately, for wide models, the decomposition tends to be uninformative. Layers are assigned none of the adversarial loss until dimensionality reduction to below the input space dimension occurs. Since the development of CNNs, state of the art architectures have used huge feature spaces. Dimensionality reduction has to occur eventually, but this methodology often results in the attribution of all the loss to the final layer.

In this paper, we propose a variation on the feature extractor distance tailored to wide networks, or equivalently feature extractors with high dimension feature spaces. We explore basic properties of this feature extractor distance and propose and implement an algorithm to approximately compute it for ReLU CNNs feature extractors. We use the algorithm to perform some initial exploration into the effects of architectural and training method decisions on layer-by-layer robustness.

As pre-trained models and the transfer learning paradigm are more widely used, analysis of these models as feature extractors rather than fully specified classifiers become more and more relevant. We illustrate the potential feasibility of the layer-by-layer approach to this task.

At this point, adversarial examples have been known to be a serious problem in neural network for over a decade (Szegedy et al., 2013; Goodfellow et al., 2015). Adversarial training has succeeded at mitigating this vulnerability to an extent, but is computationally expensive and does not achieve information-theoretic limits on adversarial robustness (Bhagoji et al., 2021). Certified methods give stronger guarantees, but are usually even more computationally expensive and limited in the scale of the adversarial budget.

We would like to provide new analysis tools to enable continued progress on training adversarially robust models. With these tools, we can better identify different behaviors in models as training method is varied. By localizing adversarial vulnerabilities within a model, we allow computational resources to be spent efficiently. Code used in our experiments is available.[1]

**Contributions.** We introduce a novel collision search algorithm that leverages gradient information to evaluate differentiable feature extractors. We provide a theoretical framework for understanding the robustness of feature extractors in high-dimensional feature spaces. We utilize random projections to transform feature extractors with high-dimensional feature spaces to representative low-dimensional versions whose robustness can be efficiently analyzed. From our experiments on convolutional networks, we find that different training methods produce significant variation in layer-by-layer robustness.

## 2 Related Work

**Robustness of Feature Extractors.** Our work primarily extends the theoretical framework established by Bhagoji et al. (2023), which links adversarial robustness to optimal transport distances. However, a significant limitation of prior work is its reliance on combinatorial search methods that scale poorly with depth. Also, it applies only to restricted activation types. We advance this framework in two key ways: First, we introduce a gradient-based collision search algorithm that evaluates robustness for any differentiable feature extractor, enabling efficient scaling to deep architectures supported by various differentiable activation functions. Second, we integrate random projections to handle high-dimensional feature spaces, overcoming the dimensionality constraints that limited previous theoretical evaluations.

**Layer-wise Analysis and Model Utilization.** Understanding how robustness and information flow evolve through network layers is an active area of research. Song et al. (2021) focused on optimizing layer-wise quantization under Lipschitz bounds, while Craig et al. (2023) derived bounds for information flow through subnetworks. More recently, Gavrikov et al. (2024) investigated how training schemes (including adversarial training) alter layer utilization, identifying which layers become critical for performance.

While Gavrikov et al. (2024) focuses on functional criticality, our work provides a *geometric* lower bound on untargeted robustness via feature collisions. Unlike prior approaches that bound Lipschitz constants

---

[1] https://github.com/Kukukloo/cnn-layerwise-robustness

or analyze parameter utilization, our method identifies concrete collision points. This dual functionality allows us to not only measure vulnerability but also generate the specific adversarial examples that exploit it, offering a direct probe into the geometry of middle-layer representations.

**Random Projections and Optimal Transport.** Our use of Gaussian random projections is philosophically rooted in the Sliced Wasserstein (SW) distance. Optimal adversarially robust classification is intimately connected to optimal transport distances between class distributions (Bhagoji et al., 2019; Pydi & Jog, 2020; Trillos et al., 2023). However, estimating these distances suffers from the curse of dimensionality. SW distance bypasses this by projecting high-dimensional measures onto lower-dimensional subspaces (Kolouri et al., 2018; Deshpande et al., 2019; Bonneel et al., 2015; Nadjahi et al., 2021). Whereas SW methods typically average 1-D distances over many slices, we employ a higher-dimensional (e.g., 10-D) random projection to approximate the optimal loss, balancing computational efficiency with the preservation of geometric structure.

## 3 Framework

In this section, we build toward the definition of our robustness measure by discussing key technical aspects of prior work. Section 3.1 characterizes the optimal adversarial cross-entropy loss in terms of a convex optimization problem with constraints described by a bipartite conflict graph. Section 3.2 explains an earlier definition of a robustness measure for feature extractors based on the optimal adversarial loss of a classifier built on that feature extractor.

We consider the following adversarial classification scenario. (For a comprehensive summary of notation, please refer to Table 1 in Appendix A.) There is an example space $\mathcal{X} = \mathbb{R}^{n_0}$ equipped with a a distance metric $d$ and a label space $\mathcal{Y} = \{0, 1, \ldots, k-1\}$. A labeled example $(x, y)$ is sampled from a distribution $P$ over $\mathcal{X} \times \mathcal{Y}$. An adversary selects a point $\tilde{x} \in \mathcal{X}$ such that $d(x, \tilde{x}) \leq \varepsilon$. The classifier receives $\tilde{x}$, uses a classifier $h$ to produce a probability vector $h(\tilde{x}) \in \mathbb{P}(\mathcal{Y})$, and incurs loss $\ell(h(\tilde{x}), y)$. The expected adversarial loss of $h$ is $\widetilde{L}(\varepsilon, P, h) = E_{(x,y) \sim P}[\sup_{\tilde{x}:d(x,\tilde{x}) \leq \varepsilon} \ell(h(\tilde{x}), y)]$.

### 3.1 Characterizing Optimal Adversarial Loss

We briefly summarize the characterization of optimal adversarial cross entropy loss for a two-class discrete data distribution Bhagoji et al. (2021).

The distance structure of the data can be summarized in a bipartite *conflict graph* $(\mathcal{V}, \mathcal{E}_\varepsilon)$ in which, the vertices $\mathcal{V}$ represent labeled examples, and the edges $\mathcal{E}_\varepsilon$ indicate the presence of intersections under a given attack budget $\epsilon$. When such an intersection exists, it signifies a fundamental limitation: no classifier can correctly classify both points in the adversarial setting. In the graph the left vertices are examples from class zero, the right vertices are examples from class one, and an edge $(x, x')$ is present when $d(x, x') \leq 2\varepsilon$. The polytope $\mathcal{P}_\epsilon \subset \mathbb{R}^V$ is defined by the following inequalities:

$$0 \leq q_v \leq 1, v \in \mathcal{V}; \quad q_v + q_{v'} \leq 1, (v, v') \in \mathcal{E}_\varepsilon.$$

For a labeled example $(x, y)$ and soft classifier $h : \mathcal{X} \to \mathbb{P}(\mathcal{Y})$, $h(x)_y$ is the probability that $h$ assigns to the correct class. Bhagoji et al. show that there is an $h$ such that $\inf_{\tilde{x}:d(x,\tilde{x}) \leq \varepsilon} h(x)_y \geq q_{(x,y)}$ for each labeled example $(x, y)$ in the support of the data distribution if and only if $q \in \mathcal{P}_\varepsilon$. Thus $\mathcal{P}_\epsilon$ is the space of achievable correct-classification-probability vectors.

Using this characterization of the correct classification vectors, we obtain a convex optimization problem for the minimum adversarial loss:

$$\widetilde{L}^*(\varepsilon, P) = \min_h \widetilde{L}(\varepsilon, P, h) = \min_{q \in \mathcal{P}_\varepsilon} \sum_{v \in \mathcal{V}} p_v \log \frac{1}{q_v}.$$

The optimal loss represents the lower bound on the loss that any possible classifier can achieve when attempting to correctly classify an adversarial example under the current perturbation.

### 3.2 A Previous Approach to Feature Extractor Robustness

This analysis has been extended to multi-layer classifiers. Consider a classifier $h : \mathbb{R}^{n_0} \to \mathbb{P}(\mathcal{Y})$ that is the composition of $\ell$ layers: $h = \text{softmax} \circ h_\ell \circ h_{\ell-1} \circ \ldots \circ h_1$, where $h_i : \mathbb{R}^{n_{i-1}} \to \mathbb{R}^{n_i}$ and $n_\ell = k = |\mathcal{Y}|$. We can consider its initial $j$ layers $f_j : R^{n_0} \to \mathbb{R}^{n_j}$, $f_j = h_j \circ \ldots \circ h_1$ as a *feature extractor*. The key idea of Bhagoji et al. is to evaluate any feature extractor $f$ via the set of all classifiers that use $f$ as their initial layers: $\mathcal{H}_f = \{g \circ f \text{ such that } g : \mathbb{R}^{n_j} \to \mathbb{P}(\mathcal{Y})\}$ (Bhagoji et al., 2023). To measure the adversarial robustness of $f$, they consider the best achievable adversarial loss over this family: $\widetilde{L}^*(\varepsilon, P, f) = \inf_{h \in \mathcal{H}_f} \widetilde{L}(\varepsilon, P, h)$.

This definition has some nice properties. As more layers are considered to be part of the feature extractor, the robustness can only decrease because $\mathcal{H}_{g \circ f} \subseteq \mathcal{H}_f$. The set of all classifiers is $\mathcal{H}_{\text{id}}$, so the identity feature extractor has perfect robustness, i.e. no reduction from input space loss lower bound.

However, there is a key weakness to the definition: *any* injective feature extractor achieves perfect adversarial robustness. A feature extractor $f : \mathbb{R}^{n_0} \to \mathbb{R}^{n_j}$ will typically be injective or close enough to only introduce a negligible amount of loss. By this definition, only feature extractors that perform significant dimensionality reduction typically cause loss of robustness.

**Colliding adversarial examples and feature layer distance**  Let $f$ be a feature extractor and let $(x, y)$ and $(x', y')$ be labeled example from different classes $(y \neq y')$. A pair of adversarial examples collide when the perturbations cause the examples to be mapped to the same point in the feature space: $\tilde{x}$ and $\tilde{x}'$ satisfy $d(x, \tilde{x}) \leq \epsilon$, $d(x', \tilde{x}') \leq \epsilon$, and $f(\tilde{x}) = f(\tilde{x}')$. This guarantees a classification error for at least one example for any backend $g$: $g(f(\tilde{x})) = g(f(\tilde{x}'))$. The minimum adversarial budget required to find inputs that collide in the feature space defines a distance between examples:

$$d_f(x, x') = \inf_{\tilde{x}, \tilde{x}' : f(\tilde{x}) = f(\tilde{x}')} \max(d(x, \tilde{x}), d(x', \tilde{x}')).$$

**Computing the Robustness Measure**  The distance $d_f$ can be used in place of the input distance $d$ to build a conflict graph and characterize optimal adversarial loss over $\mathcal{H}_f$. Define $\mathcal{E}_{f,\epsilon}$, the edge set of the conflict graph for $d_f$, analogously to $\mathcal{P}_\varepsilon$: $\mathcal{E}_{f,\epsilon} = \{(x, x') : d_f(x, x') \leq \varepsilon\}$. Define $\mathcal{P}_{f,\varepsilon} \subset \mathbb{R}^V$ is defined analogously to $\mathcal{P}_\varepsilon$, but with constraints coming from $\mathcal{E}_{f,\epsilon}$.

Bhagoji et al. show that for a discrete distributions $P$, $\widetilde{L}^*(\varepsilon, P, f)$ can be characterized similarly to $\widetilde{L}^*(\varepsilon, P)$:

$$\widetilde{L}^*(\varepsilon, P, f) = \inf_{h \in \mathcal{H}_f} \widetilde{L}(\varepsilon, P, h) = \min_{q \in \mathcal{P}_{f,\varepsilon}} \sum_{v \in V} p_v \log \frac{1}{q_v}.$$

If $\mathcal{E}_{f,\epsilon}$ is known, then $\widetilde{L}^*(\varepsilon, P, f)$ can be efficiently computed by solving a convex optimization problem.

## 4 Measuring Robustness of High-dimensional Feature Extractor

To evaluate a feature extractor $f : \mathbb{R}^{n_0} \to \mathbb{R}^{n_j}$, we consider $\widetilde{L}^*(\varepsilon, P, a \circ f)$, where $a : \mathbb{R}^{n_j} \to \mathbb{R}^m$ is a random linear function: $a(x) = Ax$, $A \in \mathbb{R}^{m \times n_j}$ is a matrix with i.i.d. standard Gaussian entries. Note that because $a$ is a random function, $d_{a \circ f}(x, x')$ and $\widetilde{L}^*(\varepsilon, P, a \circ f)$ are random variables.

**Discussion**  An injective feature extractor does not commit to any aspect of the behavior of the classifier using it: subsequent layers of the classifier could invert it. However, a feature extractor causes some classifiers extending it to have simpler representations than others. We would like to measure the robustness of a "typical" classifier extending $f$ and we use the composition of a random dimension-reducing linear function with $f$ as a proxy for this that is computationally tractable to analyze. The random linear function is comparable to a randomly initialized but untrained classifier network using the features.

### 4.1 Properties of the Robustness Measure

Now we discuss key properties of the feature extractor distance $d_{a \circ f}$ and the robustness measure $\widetilde{L}^*(\varepsilon, P, a \circ f)$.

**Invariance of $d_f$ to scaling of $f$ or $a$** For any $c \neq 0$, $\widetilde{L}^*(\varepsilon, P, cf) = \widetilde{L}^*(\varepsilon, P, f)$. A collision $Af(\tilde{x}) = Af(\tilde{x}')$ for random linear layer $A$ is still a collision after scaling by a constant. This also means that the scale of the distribution of $A$ does not affect our definitions.

**Rotation invariance** The distribution of random linear maps is rotation invariant, so any rotation of the feature space leaves all distances unchanged. When $m \leq n_j$, the random matrix $A$ has rank $m$ with probability one. It has singular value decomposition $A = U\Sigma V^T$, where $V \in \mathbb{R}^{n_j \times m}$ and $U \in \mathbb{R}^{m \times m}$ are orthogonal and $\Sigma \in \mathbb{R}^{m \times m}$ is diagonal and invertible. Then $U\Sigma V^T f(\tilde{x}) = U\Sigma V^T f(\tilde{x}')$ if and only if $V^T f(\tilde{x}) = V^T f(\tilde{x}')$. Thus using i.i.d. Gaussian matrices results in the same distance as a uniform distribution over orthogonal projections to a dimension-$m$ subspace.

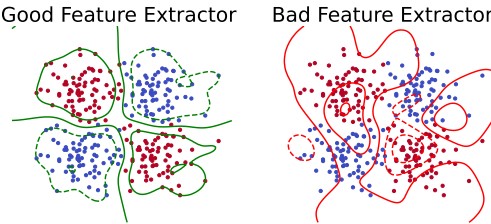

Figure 1: Random non-linear level sets for compositions of a random linear function with good and bad feature extractors. The lines represent the level sets of a random linear function composed with the feature extractor. The bad feature extractor contains all of the features in the good extractor plus many more. Most of these are low quality so a random linear combination has low robustness.

**Invertible function non-invariance** On the other hand, applying a generic invertible linear transformation $b$ to the feature space $\mathbb{R}^{n_j}$ *does* affect the distribution of the distance: it is not true that $\widetilde{L}^*(\varepsilon, P, b \circ f) = \widetilde{L}^*(\varepsilon, P, f)$. Applying the linear transformation to $f(x)$ is equivalent to applying the inverse to the random matrix $A$, which changes its distribution. All linear maps still appear, but with skewed densities.

**Adding, removing, rescaling features** While $d_f$ does not depend on the absolute scaling of features, it does depend on the relative scaling. It is easier for the adversary to satisfy $Af(\tilde{x}) = Af(\tilde{x}')$ by modifying the examples in ways that change large magnitude features.

If we modify a feature extractor $f$ to add an additional feature, giving $f'$, this could increase the distance between $x$ and $x'$ or decrease it. Adding a constant feature leaves the distance unchanged.

Figure 1 illustrates how a "good" feature extractor (where level sets are separated) can become a "bad" one (where level sets are cluttered), when enough weak features are added.

**Specializing to $\ell_2$** In the remainder of this section and in our experiments, we use $\ell_2$ distance as our input space distance.

### 4.2 Linear Feature Extractors

Suppose that $f$ is affine linear, so $f(x) = Hx + k$, and that $\mathrm{rank}(H) = r$, so the singular value decomposition of $H$ is $H = U\Sigma V^T$ where $\Sigma \in \mathbb{R}^{r \times r}$. Then Bhagoji et al. observe that the optimal adversarial examples are $\tilde{x} = x + \delta$ and $\tilde{x}' = x' - \delta$ where $\delta = \frac{1}{2} V V^T (x - x')$. The perturbation is the projection of half the difference of examples onto the orthogonal complement of the nullspace of $H$: the adversary should not allocate any of their budget to moving within the level sets of $f$. Thus $d_f(x, x')$ is $\frac{1}{2}\|V^T(x - x')\|$. The left part of Figure 2 illustrates this.

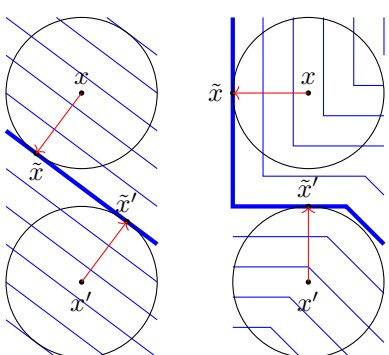

Figure 2: Visualization of $d_f(x, x')$ for two feature extractors $f : \mathbb{R}^2 \to \mathbb{R}^1$, one linear and one piecewise-linear. Blue lines are level sets of $f$. Each of the balls has radius $d_f(x, x')$. The level set of the colliding adversarial examples is highlighted. These are not classifiers and the highlighted line is not a decision boundary. When the feature space is more than one dimension smaller than the input space, level sets do not disconnect the input space.

**Proposition 1** (Informal version). *Let $f(x) = ALx$, where $L \in \mathbb{R}^{n_1 \times n_0}$ is a fixed linear feature extractor and the entries of $A \in \mathbb{R}^{m \times n_1}$ are i.i.d. standard Gaussian. Let $r_4 = \frac{\|L\|_4^4}{\|L\|_4^4}$, where $\|L\|_4$ is a Schatten p-norm and $\|L\|$ is the $\ell_2$ operator norm. Then $r_4$ is a variation of stable rank for $L$. If $1 \ll m \ll \sqrt{r_4}$, then $d_f(x, x')$ concentrates tightly around $\frac{\sqrt{m}\|L(x-x')\|}{2\|L\|_F}$.*

Precise statement and proof are in Appendix D. Note that $\|L(x - x')\|$ is just the distance between the feature representations of $x$ and $x'$. If we were only interested in linear layers, computing $d_{a \circ h}$ would not be justified: $\|L(x - x')\|$ and $\|L\|_{\mathrm{F}}$ are much more straightforward to compute, and not random! However, these norms do not obviously generalize to a nonlinear $h$, which would have different linear approximations at different points in the input space, while $d_{a \circ h}$ does still make sense. When evaluating nonlinear $h$, we would still like to pick $m$ so that $d_{a \circ h}$ concentrates so that we can efficiently and accurately estimate it by sampling.

**Input space calculation** The special case of $d_a(x, x')$ for a random linear function $a$ is more intuitive. This corresponds to $L = I \in \mathbb{R}^{n_0 \times n_0}$ and $\|L\|_{\mathrm{F}} = n_0$. The orthogonal complement of the nullspace of $A \in \mathbb{R}^{m \times n_0}$ is a uniformly distributed dimension-$m$ subspace. So $d_a(x, x')^2$ has mean $\frac{m}{4n_0}\|x - x'\|_2^2$ and $d_a(x, x')$ concentrates around $\frac{\sqrt{m}}{2\sqrt{n_0}}\|x - x'\|_2$ when $m$ is large enough.

### 4.3 Relationship between distances at different layers

Now we compare $d_{a \circ f_j}(x, x')$ and $d_{a \circ f_{j+1}}(x, x')$. As we noted before, $d_{f_j}(x, x') \geq d_{f_{j+1}}(x, x')$: adding an additional layer can only decrease distances by introducing new collisions between inputs. In contrast, $d_{a \circ f_{j+1}}$ could be either larger or smaller than $d_{a \circ f_j}$. Including an additional layer $h_{j+1}$ to the feature extractor introduces new possibilities for collisions, which reduces robustness, but it also reweights the existing feature information in a way that may reduce the effect of weak features. Figure 3 presents paired adversarial examples identified for the 7-layer model. We observe that as the feature extractor becomes deeper, the average adversarial distance $d_{a \circ f_i}(x, x')$ increases as well.

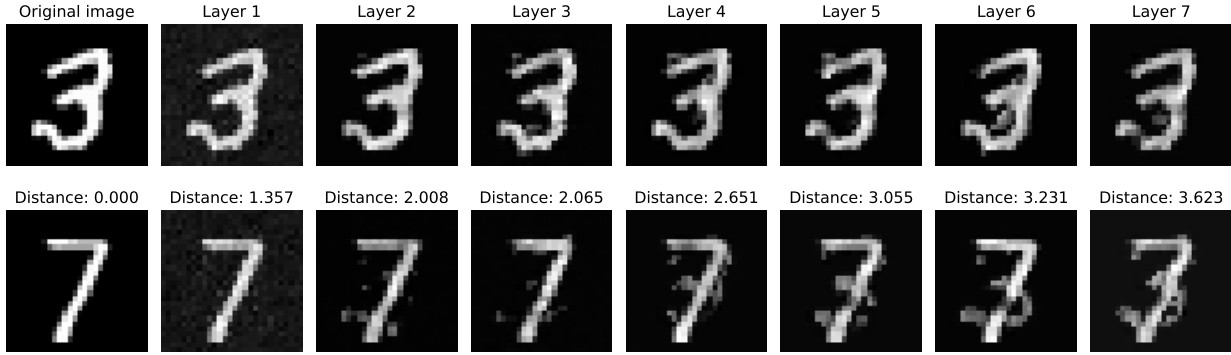

Figure 3: Colliding adversarial examples and their $\ell_2$ perturbation distance for one pair across different layers.

### 4.4 Varying the output dimension of the random projection

As we vary the output dimension $m$ of the random linear function, the distance $d_{a \circ f}$ reveal different information about the structure of the feature space representation. At one extreme, if $m \geq n_j$, then the linear map $a$ is injective with probability one and $d_f(x, x') = d_{a \circ f}(x, x')$. If $m = 1$, we roughly learn how much of the feature representation is usable by a linear classifier. By using a single fixed $m$ to analyze multiple layers with different feature space dimensions $n_i$, we eliminate some of the effects of this dimension variation on measured robustness. For most of our experiments, we use $m = 10$ to facilitate comparison with the logits.

# 5 Algorithm for approximating feature-layer distances

When we use $\ell_2$ distance to constrain our adversary, for $f : \mathbb{R}^{n_0} \to \mathbb{R}^d$, $d_f(x, x')$ becomes

$$\min_{\tilde{x}, \tilde{x}'} \|\tilde{x} - x\|_2^2 + \|\tilde{x}' - x'\|_2^2 \quad \text{subject to} \quad f(\tilde{x}) = f(\tilde{x}')$$

This is a slight relaxation: to get a smoother problem we have replaced the original objective $\max(\|\tilde{x} - x\|_2, \|\tilde{x}' - x'\|_2)$ with the sum of squares. Now we present the algorithm that we use to approximately solve this optimization problem.

Our algorithm is related to a method used to compute distance from an example to the decision boundary of a classifer. The linear approximated decision hyperplane is used in DeepFool (Moosavi-Dezfooli et al., 2016), and the FAB attack paper (Croce & Hein, 2020) employs convex combinations to make the solution closer to the original point.

---

**Algorithm 1** Collision Search Algorithm

---

**Require:** Original points $x$, $x'$; initial step $\alpha \in (0, 1)$; decay rate $\gamma \in (0, 1)$; maximum number of iteration $N_{\max}$; convergence threshold $\epsilon$.

1: Initialize $\tilde{x}^{(0)} = (x + x')/2$, $\tilde{x}'^{(0)} = (x + x')/2$, $i = 0$.
2: **for** $i = 0 \dots N_{\max} - 1$ **do**
3:      Compute Jacobians, build collision subspace:

$$\mathcal{H}_i = \{ (\tilde{x}, \tilde{x}') \in \mathbb{R}^{2n_0} \ : \ f(\tilde{x}'^{(i)}) - f(\tilde{x}^{(i)}) + Df(\tilde{x}'^{(i)})(\tilde{x}' - \tilde{x}'^{(i)}) - Df(\tilde{x}^{(i)})(\tilde{x} - \tilde{x}^{(i)}) = 0 \}$$

4:      Convex combination and projection:

$$(\tilde{x}^{(i+1)}, \tilde{x}'^{(i+1)}) = \text{proj} \left( (1 - \alpha^{(i)})(\tilde{x}^{(i)}, \tilde{x}'^{(i)}) + \alpha^{(i)}(x, x'), \mathcal{H}_i \right)$$

5:      Break if converged:

$$\left\| f(\tilde{x}'^{(i+1)}) - f(\tilde{x}^{(i+1)}) \right\| < \epsilon \quad \text{and} \quad (\|\tilde{x}^{(i)} - x\|_2^2 + \|\tilde{x}'^{(i)} - x'\|_2^2) - (\|\tilde{x}^{(i+1)} - x\|_2^2 + \|\tilde{x}'^{(i+1)} - x'\|_2^2) < \epsilon$$

6:      $\alpha^{(i+1)} = \alpha^{(i)} \times \gamma$.
7: **end for**
8: **Return** colliding adversarial examples $\tilde{x}^{(i+1)}$, $\tilde{x}'^{(i+1)}$.

---

The algorithm can be viewed as an approximate version of projected gradient descent. Because $f$ is a nonconvex piecewise-linear function, it is intractable to project onto the true constraint set $f(\tilde{x}) = f(\tilde{x}')$. Instead, we locally approximate the constraint set as an affine subspace. The derivative of $(\tilde{x}, \tilde{x}') \mapsto f(\tilde{x}) - f(\tilde{x}')$ at $(\tilde{x}, \tilde{x}')$ is the matrix $\begin{pmatrix} Df(\tilde{x}) & -Df(\tilde{x}') \end{pmatrix} \in \mathbb{R}^{m \times 2n_0}$, where $D$ represents the Jacobian matrix of the gradient. The affine subspace on which the linear approximation is zero has dimension at least $2n_0 - m$. In contrast to related algorithms to compute minimum distance to the classification boundary, the subspace usually has codimension greater than one. i.e. is not a hyperplane. Thus projection onto the subspace requires us to solve a linear system and it is preferable to choose $m$ to be not too large. Details are shown in Appendix B.

We know that $\tilde{x} = \tilde{x}'$ implies $f(\tilde{x}) = f(\tilde{x}')$, so we initialize $\tilde{x}$ and $\tilde{x}'$ by minimizing the objective subject to this constraint. Thus we set $\tilde{x}^{(0)} = \tilde{x}'^{(0)} = (x + x')/2$, the midpoint of the original examples. We found that starting with points satisfying the collision condition worked much better than any algorithm initialized at the original examples ($\tilde{x}^{(0)} = x$ and $\tilde{x}'^{(0)} = x'$).

## 5.1 Convex Combination Projection

We address the NP-hard collision-search problem (Katz et al., 2017) via a simple iterative method that alternates between convex combination and projection onto a locally linearized subspace. At iteration $i$, this subspace is $\mathcal{H}_i$. If the subspace were the true collision manifold, then the minimizer would be $\text{proj}((x, x'), \mathcal{H}_i)$. However, $\mathcal{H}_i$ is only an accurate approximation locally, so we improve the objective by mixing the projection

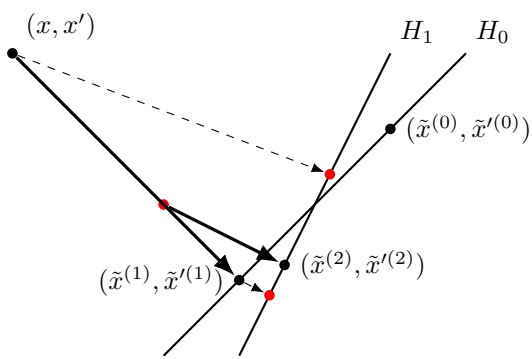

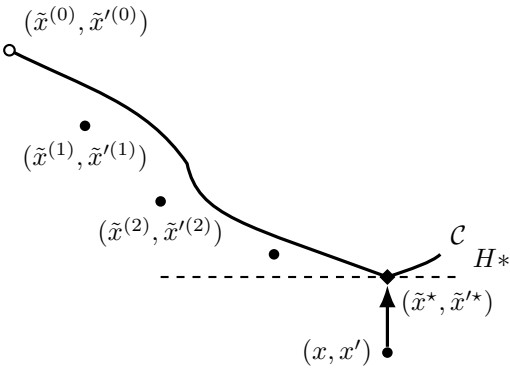

(a) First two steps. Initialize at $(x_0, x_0')$ and project orthogonally onto the first linearized collision subspace $H_0$ to obtain $(\tilde{x}^{(1)}, \tilde{x}'^{(1)})$. Relinearize to form $H_1$ and project again to get $(\tilde{x}^{(2)}, \tilde{x}'^{(2)})$. Red markers indicate the feet of the projections.

(b) Convergence of the linearize–project scheme. The true collision set $\mathcal{C}$ is curved. At iteration $i$, we linearize at $(\tilde{x}^{(i)}, \tilde{x}'^{(i)})$ to obtain the affine subspace $H_i$ and convex combination with $(x, x')$ project to $(\tilde{x}^{(i+1)}, \tilde{x}'^{(i+1)})$. The iterates approach the low point $(\tilde{x}^\star, \tilde{x}'^\star)$ on $\mathcal{C}$; $H^\star$ denotes the limiting linearization at the solution.

Figure 4: Geometry of our linearize–project algorithm on the collision manifold: (a) initialization and the first two projections; (b) iterative convergence toward the optimal point on the true collision set.

of the current estimate $(\tilde{x}^{(i)}, \tilde{x}'^{(i)})$ with the projection of original point $(x, x')$:

$$(\tilde{x}^{(i+1)}, \tilde{x}'^{(i+1)}) = (1 - \alpha^{(i)}) \operatorname{proj}\left((\tilde{x}^{(i)}, \tilde{x}'^{(i)}), \mathcal{H}_i\right) + \alpha^{(i)} \operatorname{proj}\left((x, x'), \mathcal{H}_i\right)$$
$$= \operatorname{proj}\left((1 - \alpha^{(i)})(\tilde{x}^{(i)}, \tilde{x}'^{(i)}) + \alpha^{(i)}(x, x'), \mathcal{H}_i\right)$$

The second expression for the iterate holds because the projection is affine. Using this saves computation by performing the convex combination first.

Since the midpoint $\frac{1}{2}(x + x')$ is itself a perfect collision in function space, we initialize our search there and compute its Jacobian to define the first affine subspace. In the very first projection step, we set $\alpha^{(0)} = 1$, so that $\tilde{x}^{(1)} = \operatorname{proj}(x, \mathcal{H}_0)$, using both original points to project onto $\mathcal{H}_0$. Thereafter, we gradually reduce $\alpha^{(i)}$ to introduce bias toward the original point $x$. This diminishing bias keeps every iterate within a controlled neighborhood of a feasible input, preventing large jumps that might violate constraints or break the validity of the local linear approximation. Figure 4 shows the convex-combination and projection operation in first two step as well as the convergent case in which the initial point $x$ exactly projects to the current iteration point $\tilde{x}^{(i)}$.

The algorithm proposed in Bhagoji et al. (2023) is tailored to ReLU networks: it greedily searches ReLU activation patterns to formulate a sequence of linear programs. However, as network depth increases or when non-ReLU activations are used, this combinatorial approach becomes intractable. In contrast, our Jacobian-subspace method sidesteps these limitations by relying solely on gradient information. By constructing an affine subspace from the model's local Jacobians, we can handle differentiable activation functions and deep architectures efficiently.

## 5.2 Computation and Complexity Analysis

Our algorithm alternates between (i) computing local Jacobians to linearize the collision constraint and (ii) projecting onto the resulting affine subspace.

**Time Complexity.** The computational cost to generate a collision pair scales as $O\left(N \cdot T \cdot (m \cdot \mathcal{C}_{\mathrm{bp}} + m^2 n_0)\right)$, where $N$ is the number of random projections, $T$ is the number of iterations, $n_0$ is the input dimension, and $m$ is the projection dimension (default $m = 10$). Here, $\mathcal{C}_{\mathrm{bp}}$ represents the cost of a standard backpropagation

pass through the feature extractor. The term $m \cdot \mathcal{C}_{\mathrm{bp}}$ reflects the computation of the Jacobian (via $m$ vector-Jacobian products), while $m^2 n_0$ accounts for solving the linear system for the projection. In practice, with small $m$, the runtime is dominated by the backpropagation steps.

**Memory Complexity.** Peak memory usage is determined by the storage of intermediate activations required for gradient computation. This scales linearly with $O(\text{Batch} \times \text{Depth} \times \text{Resolution})$. For our 7-layer CNN on CIFAR-10, we can process batches of 100 pairs on a 40GB GPU; however, for deeper architectures like WideResNet or higher-resolution inputs, the batch size must be reduced (e.g., to $\sim 15$) to remain within VRAM limits.

# 6 Experiments

In this section, we test several theories and evaluate the robustness of convolutional models layer by layer on the MNIST (LeCun et al., 1998) and CIFAR-10 (Krizhevsky & Hinton, 2009) datasets. Since MNIST is a relatively simple dataset, it allows us to easily explore robustness properties. And the MNIST dataset highlights a larger robustness gap between layers and provides a range of $\epsilon$ values for testing. For our baseline, we use an adversarially trained model on MNIST with projected gradient descent (PGD) (Madry et al., 2017) and set perturbation budget $\epsilon = 2$.

To design the experiment, we selected 100 examples from two classes in the training dataset for evaluation. For the robustness analysis, we focus on fixed binary class pairs to keep the setup simple and reproducible. On MNIST and Cifar-10 we use class 3 vs. 7. For each dataset we randomly sample $S = 100$ training examples per class and evaluate our metric on *all* cross-class pairs, resulting in $S^2 = 10000$ pairs. Since our collision-search algorithm is run on every pair and for multiple random projections, the overall cost scales quadratically in $S$, and larger subsets quickly become computationally prohibitive. In Appendix Section C, we show that increasing the subset to $S = 200$ and changing the class pair or the sampled subset yields qualitatively similar layer-wise robustness curves, indicating that our conclusions are robust to these choices.

---

**Algorithm 2** Optimal Loss Calculation via Conflict Graph

---

1: Choose feature extractor $f_j : \mathbb{R}^{n_0} \to \mathbb{R}^{n_j}$
2: Specify the number of random projections $N$, and projection dimension $m$
3: **for** $i = 1$ to $N$ **do**
4:     Generate random projection matrix $A_i \in \mathbb{R}^{m \times n_j}$, with elements drawn i.i.d. from $\mathcal{N}(0,1)$
5:     Build new feature extractor $(a_i \circ f_j)(x) = A_i f_j(x)$
6:     **for** example pairs $(x, x')$ **do**
7:         Approximate $d_{a_i \circ f_j}(x, x')$ using Algorithm 1,
8:     **end for**
9:     **for** $\varepsilon = \varepsilon_{\min} \ldots \varepsilon_{\max}$ **do**
10:         Build conflict graph $(V, E_\varepsilon)$, where:
11:            $V$ is the set of data, $x, x' \in V$
12:            $E_\varepsilon = \{(x, x') : d_{a_i \circ f_j}(x, x') \le \varepsilon\}$
13:         Find $L_{\mathrm{opt}}(\varepsilon)$ using Algorithm 1 from Bhagoji et al. (2021)
14:     **end for**
15: **end for**
16: **Return** List of averaged optimal losses $L_{\mathrm{opt}}$

---

For the selection of CNN models, we trained three different architectures, following the work in (Gowal et al., 2019), where all layers utilize ReLU activation functions. We employed two robust training methods: PGD adversarial training, and TRADES adversarial training (Zhang et al., 2019). Each of these methods was applied with robustness constraints based on the $l_2$ norm. For all models, we used a cosine learning rate schedule (Loshchilov & Hutter, 2016) with an initial learning rate set to 0.01 to ensure consistency across training approaches.

**Optimal Loss** We use optimal loss $\widetilde{L}^*(\varepsilon, P, a \circ f)$ defined in Section 3.2 to represent the robustness of the feature extractor. Recall that higher loss represents lower robustness performance. Algorithm 2 computes

the optimal loss using a conflict graph–based procedure. The process begins by selecting a feature extractor and generating a sequence of random matrices with i.i.d. Gaussian entries. Each random matrix $A_i$ is combined with the feature extractor $f_j$ to create a lower-dimensional feature mapping $x \mapsto A_i f_j(x)$, making the representation suitable for robustness analysis. The algorithm then evaluates all example pairs to estimate pairwise distances and identifies collisions according to a perturbation-based threshold $\varepsilon$. These collisions define the edges of a conflict graph, where nodes represent examples and edges connect pairs within the specified perturbation distance. For each adversarial budget the algorithm invokes Algorithm 1 from Bhagoji et al. (2021) to compute the optimal loss over the conflict graph. Finally, the results are averaged over multiple random matrices to produce the final optimal loss estimates.

## 6.1 Validating the robustness metric

To evaluate the true robustness loss on both the training and test datasets, we additionally compare our collision-based loss to the FAB attack, which provides a standard decision-boundary–based robustness metric, rather than relying only on the training-time attack loss. FAB computes, for a single input, the minimum perturbation required to change its predicted label, whereas our collision-based procedure at the logit layer jointly moves a pair $(x, x')$ from different classes to a point where their logits coincide.

In our setup, the worst-case loss is $\log(2)$, indicating that even when collisions occur, we still have a 50% chance of correctly classifying a randomly chosen element of the pair. As shown in the figure 5a, when the value of $\epsilon$ increases, the adversarial loss measured by FAB rises, while our loss saturates at its maximum value $\log(2)$. Thus, the FAB loss remains consistently higher than our collision-based loss, which is expected because FAB directly optimizes misclassification of a single input, whereas our metric is designed to capture how easily feature representations of different classes can be brought to coincide.

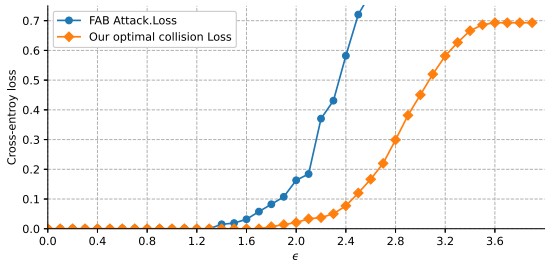
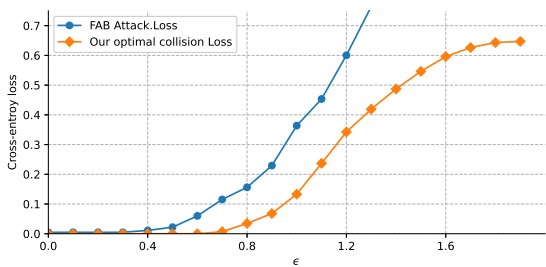

(a) MNIST Medium CNN (PGD-2).      (b) CIFAR-10 Medium CNN (PGD-0.5).

Figure 5: Comparison of FAB Attack Loss vs. Our Optimal Collision Loss.

## 6.2 Varying the dimension of the projected features

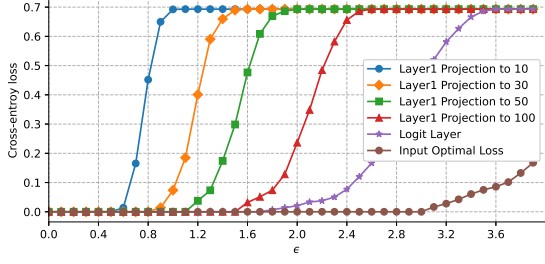
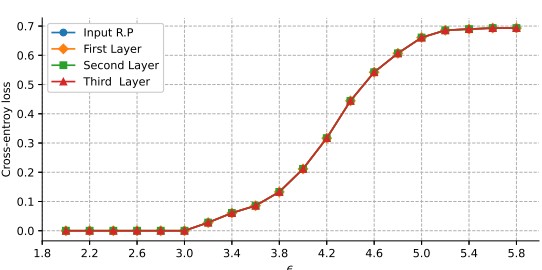

Figure 6: MNIST Medium CNN (PGD-2): First conv layer with varying random projection dimensions ($m \to 784$).

Figure 7: Medium CNN optimal loss using the method in Bhagoji et al. (2023).

Since the output dimension of the convolution is higher than the input, when no random projection is applied, the only collisions occur when the pairs are identical. Therefore, the distance between pairs reflects the exact

input distance, as shown by the brown line in the figure 6. When projecting to a higher-dimensional space, we introduce less noise while preserving more useful feature information. As a result, the dimension increase, and the loss decreases.

The distances scale roughly with $\sqrt{m}$, but not exactly, which reflects the somewhat different information about feature structure accessed by different choices of $m$.

### 6.3 Baseline layer-by-layer robustness results

When applying the method of Bhagoji et al. (2023), the search algorithm cannot find any nontrivial collisions because none of the initial linear layers have an output dimension smaller than the example space dimension. The robustness measure for these initial layers remains essentially the same as at the input level. This is illustrated in 7. Layer dimensions for all models investigate are listed in Table 3 in Appendix C.

In Figures 8 and 9, we have baseline layer-by-layer results for MNIST and CIFAR10 respectively. As we move through the layers, robustness increases, suggesting that low-quality features are being pruned. This effect is far outweighing any reductions in robustness due to the introduction of new opportunities for collisions as the feature dimension spaces decrease. Clearly, the dimensions of the feature spaces have a significant impact on the robustness measure, but they do not explain the curves by themselves. For both datasets, there is a relatively large improvement from layer one to layer two and from layer four to layer five. These layers perform dimensionality reduction by a factors of four and six, which are larger than reductions in adjacent layers. However, from layer six to the logits, we do not observe much robustness improvement despite dimensionality reduction from 512 to 10.

As we move through the layers, we observe that the optimal loss decreases, indicating that the model gradually learns more about the data's robustness, and the features become increasingly separable. Notably, there is a distinct gap between the convolutional layers and the fully connected layers. Additionally, the random projection appears to perform similarly to the learned matrix. The last few linear layers, however, seem to contribute little to the overall robustness.

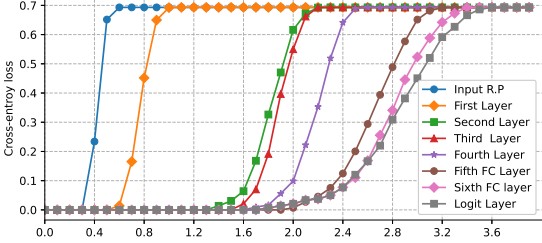

Figure 8: MNIST Medium CNN Model trained with PGD-2: All layers robustness.

Figure 9: CIFAR-10 Medium CNN Model trained with PGD-0.5: All layer robustness.

### 6.4 Effects of architecture on layer-by-layer robustness

When comparing the layer-wise robustness of the small CNN in figure 10 and the Large CNN in figure 11 on MNIST, we observe that robustness does not change uniformly throughout the models. The same observation shows in robustness on CIFAR in figure 12. Although the small CNN exhibits lower overall robustness, its first convolutional layer appears to perform better, and its second layer achieves a similar level of robustness to that of the medium CNN.

### 6.5 Effects of training method on layer-by-layer robustness

In Figures 13,14,15, we examine the medium CNN models trained with methods other than plain PGD. These can be compared to Figure 8. TRADES have layer-one robustness cliffs at lower adversarial budgets than

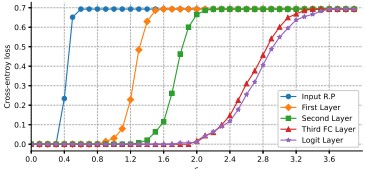

Figure 10: MNIST Small CNN Model trained with PGD-2: All layer robustness.

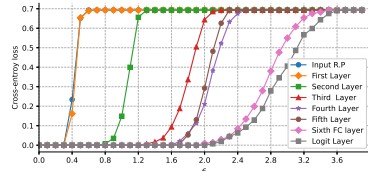

Figure 11: MNIST Large CNN Model trained with PGD-2: All layer robustness.

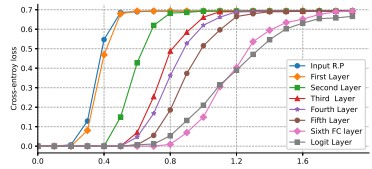

Figure 12: CIFAR-10 Large CNN Model trained with PGD-0.5: All layer robustness.

PGD. Besides, TRADES achieves more separation between fully-connected layer curves and more gradual growth of those curves.

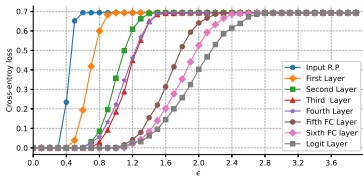

Figure 13: MNIST Medium CNN Model with benign training: All layer robustness.

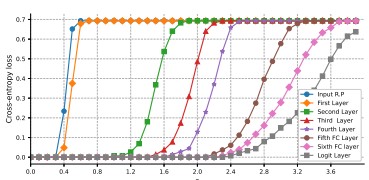

Figure 14: MNIST Medium CNN Model trained with TRADES-2: All layer robustness.

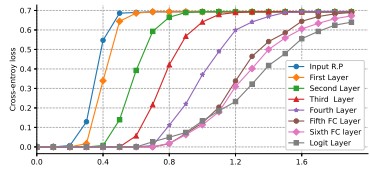

Figure 15: CIFAR-10 Medium CNN Model trained with TRADES-0.5: All layer robustness.

**Relationship between intermediate layer robustness and test robust accuracy** In Figures 17a and 17b, we see the test robust accuracy of the models. As expected, benign training achieves very limited robustness, TRADES significantly outperforms plain PGD. It is unclear to us whether the layer-one behavior of TRADES should be viewed as an undesirable side-effect that suggests room for improvement, or an indication that layer-one robustness curves are not appropriate objectives.

### 6.6 Conclusion

An interesting finding is that, for models exhibiting greater robustness under the norm-based attack, the optimal loss at the logit layer remains relatively low, which indicates high overall robustness, yet the first convolutional layer does not consistently follow this trend. In fact, this layer often learns less robust features even when the model as a whole is quite robust. This suggests that prioritizing the robustness of the first convolutional layer early in the training process is a potential avenue for improving a model's robustness.

At the same time, our experiments show that robustness does not grow uniformly across layers or blocks or depend on feature space dimension a simple manner. Some adjacent layers have similar robustness, and robustness levels often stabilize before the final linear layer. Of course, this behavior varies with different datasets, training methods, and architectures.

### 6.7 Limitations and Future Work.

Our method requires Jacobian computations for input pairs, imposing a memory overhead that scales linearly with input resolution and network depth. This trade-off necessitates smaller batch sizes for very deep or high-resolution models compared to standard training. Additionally, while our collision search is algorithmically compatible with various constraints, our current theoretical concentration bounds are derived for the rotationally invariant $l_2$ geometry. Developing rigorous theoretical guarantees for the $l_\infty$ norm involves addressing non-trivial challenges regarding the geometry of intersection sets in high dimensions, which constitutes a key direction for our future work.

**Broader Impact Statement**

Analysis of feature extractor robustness can be used to detect weaknesses in classifiers and localize these to particular layers. This is intended to serve the goal of building more robust and trustworthy models and is likely to be most useful for these purposes. For example, weak layers can be retrained or rearchitected. However, there is some potential for any method of assessing robustness to also be useful for the design and implementation of attacks: locating weaknesses in classifiers so they can be exploited rather than fixed. The potential for misuse is mitigated by the fact that our collision finding algorithm has been designed for use in the white-box setting.

**Acknowledgments**

This research used the Delta advanced computing and data resource which is supported by the National Science Foundation (award OAC 2005572) and the State of Illinois. Delta is a joint effort of the University of Illinois Urbana-Champaign and its National Center for Supercomputing Applications.

This work used Delta CPU and Delta GPU at NCSA through allocation CIS240583 from the Advanced Cyberinfrastructure Coordination Ecosystem: Services & Support (ACCESS) program, which is supported by U.S. National Science Foundation grants #2138259, #2138286, #2138307, #2137603, and #2138296. (Boerner et al., 2023)

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

## A   Symbol Description

Table 1: Summary of commonly used notation.

| Symbol | Description |
|---|---|
| $\mathcal{X} \subseteq \mathbb{R}^{n_0}$ | Input space with dimension $n_0$. |
| $\mathcal{Y} = \{0, \ldots, k-1\}$ | Label space for $k$ classes. |
| $(x, y) \sim P$ | A labeled example drawn from data distribution $P$. |
| $d(x, x')$ | Distance metric in the input space (typically $\ell_2$). |
| $\epsilon$ | Adversarial perturbation budget. |
| $f : \mathbb{R}^{n_0} \to \mathbb{R}^{n_j}$ | A feature extractor comprising the first $j$ layers of a network. |
| $h : \mathcal{X} \to \Delta^{k-1}$ | A full classifier mapping inputs to class probabilities. |
| $\mathcal{H}_f$ | The set of all classifiers that use $f$ as their feature extractor. |
| $A \in \mathbb{R}^{m \times n_j}$ | A random projection matrix with i.i.d. Gaussian entries. |
| $a : \mathbb{R}^{n_j} \to \mathbb{R}^m$ | Random linear projection function, defined as $a(z) = Az$. |
| $d_f(x, x')$ | The minimum input distance required to cause a collision in feature space $f$. |
| $d_{a \circ f}(x, x')$ | The feature-layer distance after random projection. |
| $\tilde{L}^*(\epsilon, P, f)$ | The optimal adversarial loss (lower bound) achievable by any classifier using feature extractor $f$. |

## B   Distance Calculation

### B.1   Projection operation

The projection onto a collision affine subspace in the $l_2$ norm can be formulated as the following optimization problem:

$$\min_{\delta, \delta'} \left\| \begin{bmatrix} \delta \\ \delta' \end{bmatrix} \right\|_2$$

Subject to:

$$\begin{bmatrix} Df(x^i), & -Df(x'^i) \end{bmatrix} \begin{bmatrix} \delta \\ \delta' \end{bmatrix} = f(x'^i) - f(x^i)$$

In practice, we solve this optimization problem using PyTorch's least squares solver in code.

For the first linear layer, the problem of finding the distance can be framed as solving the following optimization problem:

$$\min_{\delta, \delta'} \left\| \begin{bmatrix} \delta \\ \delta' \end{bmatrix} \right\|_2$$

Subject to:

$$AL(x + \delta) = AL(x' + \delta')$$

where A is a random projection matrix, and L represents the weights of the first linear layer. The optimal $\delta$ is the projection of $(x - x')/2$ onto the orthogonal complement of the nullspace of $AL$. Assuming $AL$ has full row rank,

$$\|\delta\|_2^2 = \frac{1}{4}(x - x')^T A^T L^T (LAA^T L^T)^{-1} AL(x - x').$$

When $A \in \mathbb{R}^{1 \times n_1}$ (so it is a single row drawn from an i.i.d. Gaussian distribution), This simplifies

$$\|\delta\|_2^2 = \frac{\|AL(x - x')\|^2}{4\|AL\|^2} = \frac{\|B\Sigma w\|^2}{4\|B\Sigma\|^2} = \frac{\sum_{i=0}^{n_1 - 1}(b_i \sigma_i w_i)^2}{4 \sum_{i=0}^{n_1 - 1}(b_i \sigma_i)^2}$$

where $b_i$ are random Gaussians, $\sigma_i$ are singular values of $L$, and $w_i$ are the coefficients of $(x - x')$ is the basis of right singular vectors. Thus $w_i$ measures the usefulness of a particular singular vector for distinguishing $x$ from $x'$. Weak features contribute proportionally more to the denominator. By assigning these smaller singular values (or even zero), $\|\delta\|$ could be increased.

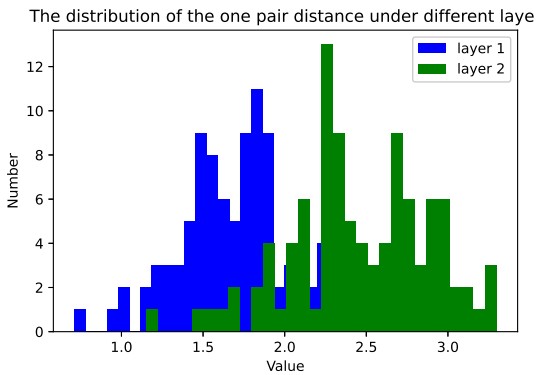

Figure 16: One pair distance under different random projection matrix

## B.2  Concentration

We measured the distance between one pair in the first and second layers with 100 different random gaussian projection matrix, and the distribution of these distances is presented in the histogram. The figure 16 shows that the distances are concentrated, clearly highlighting the differing distributions between the two layers.

## C  Experiments

### C.1  Robust accuracy of the models

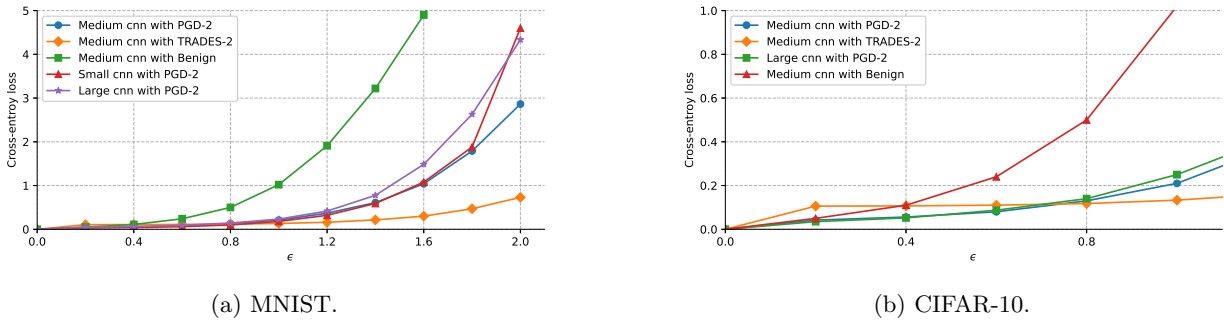

(a) MNIST.                                          (b) CIFAR-10.
Figure 17: Comparison of AutoAttack test loss for MNIST and CIFAR-10 models.

## C.2  Runtime and Convergence

To demonstrate the computational feasibility of our approach, we report the wall-clock runtimes for generating collision pairs in Table 2. Measurements were conducted on a single NVIDIA A100 GPU.

As shown in the table, the computational cost increases marginally with network depth, reflecting the accumulation of backpropagation steps required to compute the Jacobian for deeper feature layers. However,

even for the final layers, the runtime remains approximately 1 second per batch of 100 pairs, confirming that the method is efficient enough for extensive layer-wise analysis.

Regarding the stopping criterion for Algorithm 1, we employ a strict convergence threshold to ensure the stability of the found collisions. The search terminates when the update magnitude for the adversarial pair becomes negligible between iterations $t$ and $t+1$:

$$\|\tilde{x}^{(t+1)} - \tilde{x}^{(t)}\|_2 + \|\tilde{x}'^{(t+1)} - \tilde{x}'^{(t)}\|_2 < 10^{-2}.$$

| Dataset | Model | Res. | Batch | Layer Time (seconds) | | | | | | |
|---|---|---|---|---|---|---|---|---|---|---|
| | | | | L1 | L2 | L3 | L4 | L5 | L6 | L7 |
| MNIST | Medium-CNN | $28^2$ | 100 | 0.66 | 0.74 | 0.76 | 0.77 | 0.76 | 0.78 | 0.82 |
| CIFAR-10 | Medium-CNN | $32^2$ | 100 | 0.82 | 0.90 | 0.95 | 1.04 | 1.02 | 1.03 | 1.06 |

Table 2: Average wall-clock time to generate a batch of collision pairs (100 pairs) across layers L1–L7.

## C.3  Standard Deviation

For two experiments, we have added error bars representing the standard deviation due to the random projection matrix for 7 layer medium CNN model. This was found by repeating with 15 random projection matrices. Figure 18 shows that the variance is low, confirming the metric's stability.

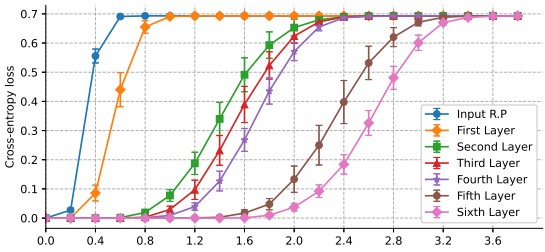

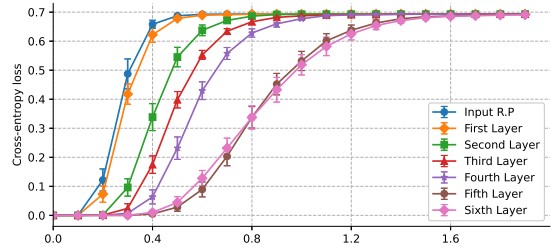

(a) MNIST Medium CNN PGD-2: Standard deviation across 15 projections.

(b) CIFAR-10 Medium CNN PGD- 0.5: Standard deviation across 15 projections.

Figure 18: **Stability Analysis.** The proposed optimal loss curves show very low variance across different random projection initializations, confirming the reliability of the metric.

## C.4  Activation function and batchnorm

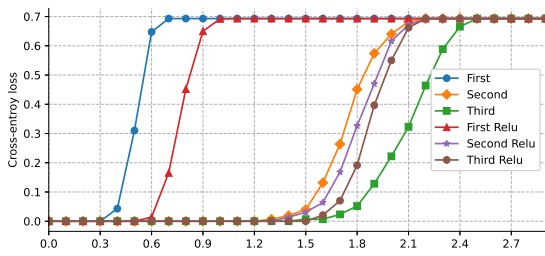

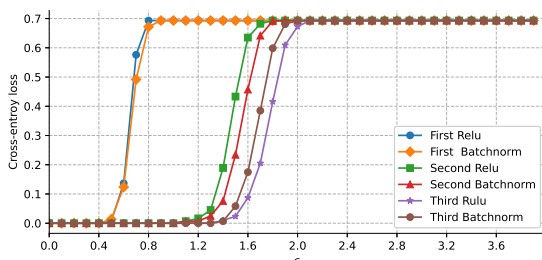

Figure 19: The ReLU performance

Figure 20: The Batchnorm performance

We evaluated the effect of including the ReLU activation functions as part of the feature extractor. As the ReLU is non-injective, it enables new types of input collisions and could potentially reduce robustness because of this. On the other hand, ReLUs can zero out weak features and improve robustness. The results in figure 19 indicate that the ReLU activation in the first and second layers effectively reduces the number of collisions,

| Model | Layer Description | Out Dim. MNIST | Out Dim. CIFAR |
|---|---|---|---|
| Med. CNN | Conv1: 32, $3 \times 3$, stride 1 | 25088 | 32768 |
| | Conv2: 32, $4 \times 4$, stride 2 | 6272 | 8192 |
| | Conv3: 64, $3 \times 3$, stride 1 | 12544 | 16384 |
| | Conv4: 64, $4 \times 4$, stride 2 | 3136 | 4096 |
| | FC: 512 | 512 | 512 |
| | FC: 512 | 512 | 512 |
| | FC: 10 | 10 | 10 |
| Small CNN | Conv1: 16, $4 \times 4$, stride 2 | 3136 | 4096 |
| | Conv2: 32, $4 \times 4$, stride 1 | 5408 | 7200 |
| | FC: 100 | 100 | 100 |
| | FC: 10 | 10 | 10 |
| Large CNN | Conv1: 64, $3 \times 3$, stride 1 | 50176 | 65536 |
| | Conv2: 64, $3 \times 3$, stride 1 | 50176 | 65536 |
| | Conv3: 128, $3 \times 3$, stride 2 | 25088 | 32768 |
| | Conv4: 128, $3 \times 3$, stride 1 | 25088 | 32768 |
| | Conv5: 128, $3 \times 3$, stride 1 | 25088 | 32768 |
| | FC: 512 | 512 | 512 |
| | FC: 10 | 10 | 10 |
| AlexNet | Conv1: 64, $5 \times 5$, stride 1, MP3 | 12544 | 16384 |
| | Conv2: 192, $5 \times 5$, stride 1, MP3 | 9408 | 12288 |
| | Conv3: 384, $3 \times 3$, stride 1, MP3 | 18816 | 24576 |
| | Conv4: 256, $3 \times 3$, stride 1, MP3 | 12544 | 16384 |
| | Conv5: 256, $3 \times 3$, stride 1, MP3 | 2304 | 4096 |
| | FC: 4096 | 4096 | 4096 |
| | FC: 4096 | 4096 | 4096 |
| | FC: 10 | 10 | 10 |
| Wide ResNet 28-10 | Block 1: Res, $3 \times 3$, 160 filters | 125440 | 163840 |
| | Block 2: Res, $3 \times 3$, 160 filters | 125440 | 163840 |
| | Block 4: Res, $3 \times 3$, 320 filters | 125440 | 163840 |
| | Block 6: Res, $3 \times 3$, 320 filters | 62720 | 81920 |
| | Block 8: Res, $3 \times 3$, 640 filters | 62720 | 81920 |
| | Block 10: Res, $3 \times 3$, 640 filters | 31360 | 40960 |
| | Block 12: Res, $3 \times 3$, 640 filters | 31360 | 40960 |
| | FC: 10 | 10 | 10 |

Table 3: Model Architectures and Feature Space Dimensions for MNIST and CIFAR-10.

suggesting a lower optimal loss in these layers. However, the situation changes with the third ReLU layer, where the activation function leads to an increase in collisions.

A similar pattern was observed with the batch normalization (BatchNorm) layers in figure 20. Adding BatchNorm after each layer showed that the first and second BatchNorm layers contributed to reduced collisions, whereas the third BatchNorm layer resulted in more collisions.

### C.5    Different activation function

We run experiments using **GELU** and **Tanh** activations on MNIST. As shown in Figure 21 (included in the revised appendix), these experiments yield qualitatively similar robustness curves and convergence behaviors to the ReLU models. This supports our claim that the collision-search method is not strictly tied to piecewise-linear activations like ReLU.

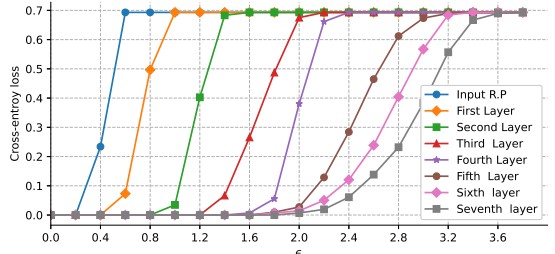
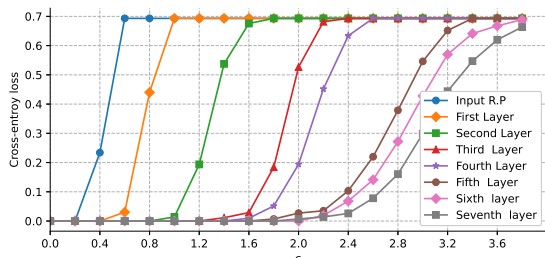

(a) The optimal loss for the MNIST model with Gelu activation function.

(b) The optimal loss for the MNIST model with Tanh activation function.

Figure 21: Comparison of optimal loss curves for different activation functions.

### C.6    Impact of sample size and class choice.

To assess robustness of the conclusions, we perform a sensitivity analysis on the Medium CNN. First, we increase the subset size to $S = 200$ per class (40,000 pairs). Second, we repeat the analysis with an alternative MNIST class pair (2 vs. 5). Third, we re-sample a different set of $S = 100$ examples for the original class pair. As shown in Fig. 22, the layer-wise robustness curves and the relative ordering of layers remain qualitatively unchanged across these variations, indicating that our conclusions are stable with respect to sample size and class choice.

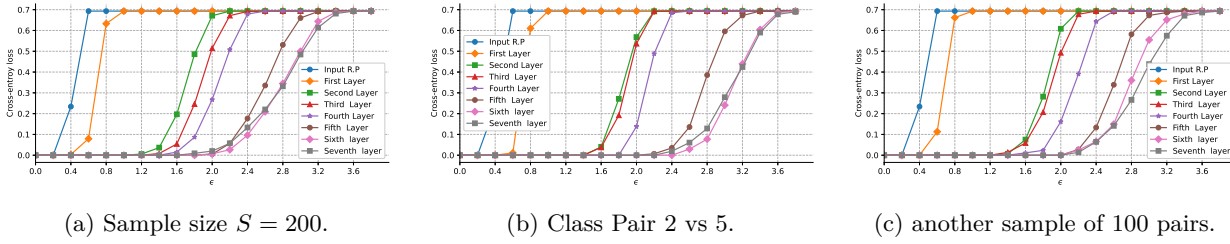

(a) Sample size $S = 200$.

(b) Class Pair 2 vs 5.

(c) another sample of 100 pairs.

Figure 22: Sensitivity Analysis. The layer-wise robustness trends remain consistent when (a) increasing sample size to 200, (b) changing the class pair, and (c) another 100 pairs.

### C.7    Different learning rate

To investigate how the learning rate affects layer-wise robustness, we retrained the model using a flat learning rate instead of a cosine schedule. At similar learning rate levels, the model trained with a flat learning rate exhibits no significant difference in layer-wise robustness compared to the one trained with a cosine schedule. However, when the learning rate is reduced to 0.001 in figure 24, the optimal loss curve narrows starting from

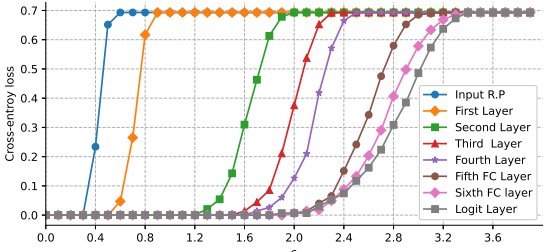

Figure 23: MNIST Medium Cnn Model trained with PGD: All layer robustness, learning rate is 0.01 throughout training

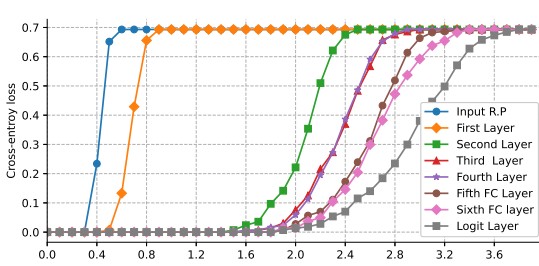

Figure 24: MNIST Medium Cnn Model trained with PGD: All layer robustness, learning rate is 0.001 throughout training

the second layer. This indicates that the model achieves good robustness with just the first two layers, while the subsequent layers contribute only marginally to robustness.

## C.8 Changing the width of a single layer

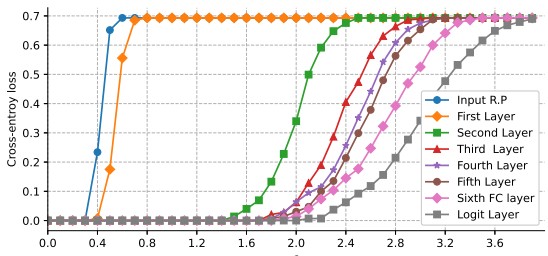

Figure 25: MNIST Medium Cnn Model trained with PGD: All layer robustness, and the first convolutional layer has more filters

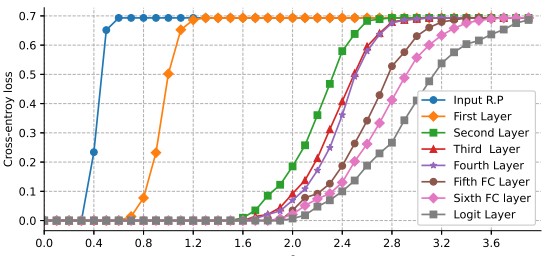

Figure 26: MNIST Medium Cnn Model trained with PGD: All layer robustness, and the first convolutional layer has fewer filters

When we decrease the number of filters in the third layer, the overall robustness of the model slightly decreases in figure 26 and figure 25, but the robustness of the first layer improves, which aligns with our conclusion in the main paper. Conversely, increasing the number of filters in the first layer leads to the opposite effect, with a slight decrease in first-layer robustness and a potential increase in overall robustness.

## C.9 Robust training with varying perturbation budgets

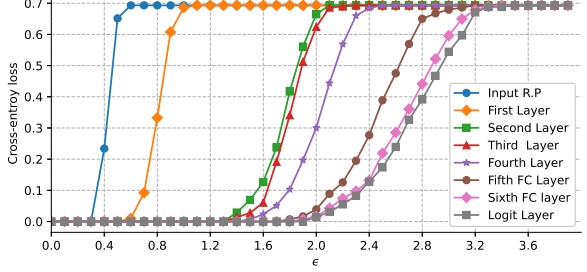

Figure 27: MNIST Medium Cnn Model trained with PGD-1: All layer robustness

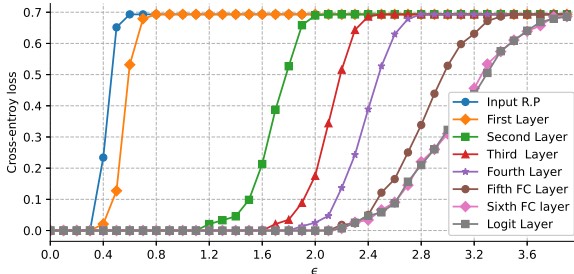

Figure 28: MNIST Medium Cnn Model trained with PGD-3: All layer robustness,

When training with different perturbation budgets, we observed that, compared to PGD-2, PGD-1 exhibits similar robustness in the first and second layers. However, in the remaining layers, more collisions are detected,

indicating lower robustness as reflected by the higher optimal loss. In contrast, PGD-3 demonstrates greater robustness than PGD-2, with the third layer showing improved distance and lower optimal loss compared to the second layer—a pattern not observed in PGD-2. The robustness distance in the other layers follows the same properties previously identified.

## C.10 More common architecture

WideResNet-28 has three stages, each consisting of four blocks, for a total of twelve blocks. AlexNet, on the other hand, includes five convolutional layers with max pooling and three fully connected layers. The feature dimensions are shown in Table 3.

For MNIST, the performance of AlexNet aligns with the observations reported for other medium-sized CNN models in the paper. However, when evaluating CIFAR, it appears that the max pooling operation concentrates robustness in the layer following the third pool.

In the case of WideResNet-28 on MNIST, we observe that robustness information is gradually learned by each block. However, for CIFAR, the final linear layer retains the most robustness-related information. While the robustness of individual blocks remains similar, the final linear layer enhances overall robustness, bringing it to a level comparable with other models.

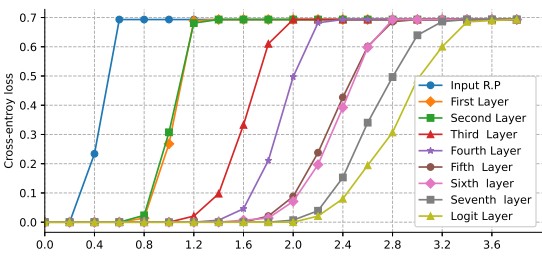

Figure 29: AlexNet Robustness on MNIST

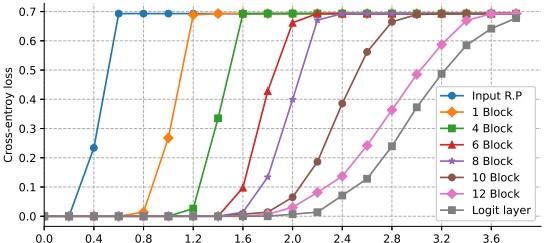

Figure 30: The WideResNet PGD-2 robustness on Mnist

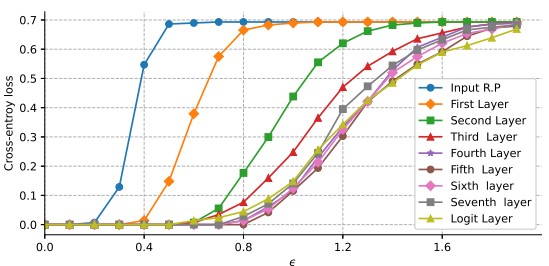

Figure 31: AlexNet Robustness on CIFAR

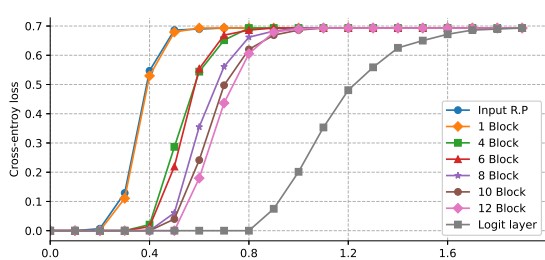

Figure 32: The WideResNet PGD-0.5 robustness on CIFAR

## C.11 PCA

Here, we used PCA instead of training an actual model, and then performed random projection on the PCA down to 10 dimensions for comparison. This illustrates how removing low-quality features can improve robustness as we measure it. The PCA on CIFAR-10 retains much greater distance compared to a real model.

## C.12 Using feature extractor for training

To show that feature extractors really matter for robustness, we run a simple freeze-$k$ test: freeze the first $k$ layers ($k = 0 \ldots 7$), re-initialize the rest, do one epoch of PGD-2 adversarial training on the unfrozen part,

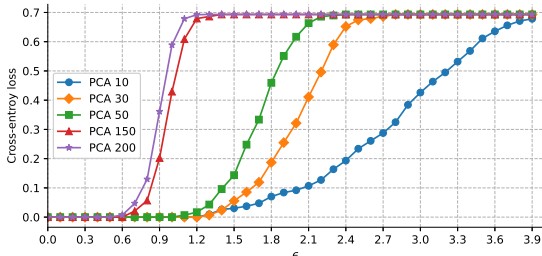

Figure 33: Mnist PCA, then random projection to 10

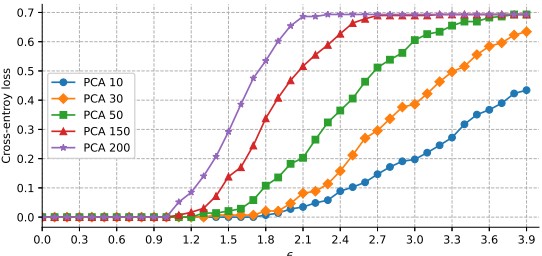

Figure 34: Cifar10 PCA, then random projection to 10

and record the adversarial training loss for $\epsilon \in [2.0, 3.0]$. Lower loss means the frozen extractor already makes robust learning easier for the new head.

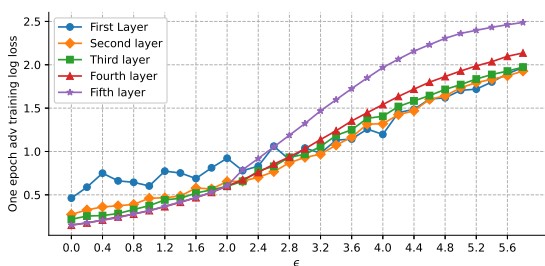

Figure 35: Mnist benign model one epoch

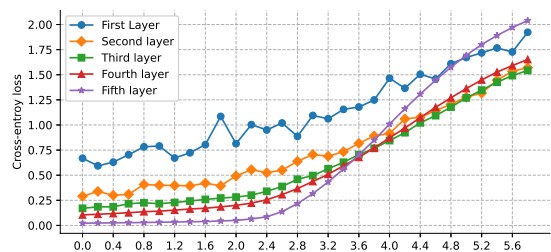

Figure 36: Mnist adv trained on PGD-2 model one epoch

For the clean-trained ("benign") model, the one-epoch adversarial loss increases as $k$ grows, showing that fixing more clean features hurts robust adaptation. For the adversarially pre-trained ("robust") model, the loss decreases with $k$, indicating that its features provide a stable, robustness-friendly backbone. This freeze-$k$ test is an external sanity check that different feature extractors change downstream robust training. Our contribution is a fast, training-free test that scores extractor robustness. In experiments, our score aligns with the freeze-$k$ behavior, so practitioners can pick good extractors without re-training many models.

## D  Analysis of the distance for linear layers

Throughout, let $r_4 = \frac{\|L\|_4^4}{\|L\|^4}$, where $\|L\|_4$ is a Schatten $p$-norm and $\|L\|$ is the $\ell_2$ operator norm. Then $r_4$ is a variation of stable rank for $L$.

**Proposition 2.** *Let $f(x) = ALx$, where $L \in \mathbb{R}^{n_1 \times n_0}$ is a fixed linear feature extractor and the entries of $A \in \mathbb{R}^{m \times n_1}$ are i.i.d. standard Gaussian. For $\epsilon \geq 2\max(e^{-m/4}, m^2 e^{-r_4})$,*

$$\frac{1 - \frac{2}{\sqrt{m}}\sqrt{\log(2/\epsilon)}}{1 + \frac{4m\|L\|_4^2}{\|L\|_F^2}\sqrt{\log(m^2/\epsilon)}} \leq \frac{d(x,x')^2}{\frac{m\|L(x-x')\|^2}{4\|L\|_F^2}} \leq \frac{1 + \frac{4}{\sqrt{m}}\sqrt{\log(2/\epsilon)}}{1 - \frac{4m\|L\|_4^2}{\|L\|_F^2}\sqrt{\log(m^2/\epsilon)}}$$

*with probability $1 - \epsilon$.*

This bound implies concentration of the distance when $m \gg 1$ and $m \ll \frac{\|\sigma\|_2^2}{\|\sigma\|_4^2} = \frac{\|\sigma\|_2^2}{\|\sigma\|_\infty^2} \frac{\|\sigma\|_\infty^2}{\|\sigma\|_4^2} = \frac{r_2}{\sqrt{r_4}}$. Because $r_4 = \sum_k \left(\frac{\sigma_k}{\sigma_1}\right)^4 \leq \sum_k \left(\frac{\sigma_k}{\sigma_1}\right)^2 = r_2$, $\sqrt{r_4} \leq \frac{r_2}{\sqrt{r_4}}$ and $m \ll \sqrt{r_4}$ is sufficient to get concentration.

*Proof.* For $f(x) = Hx$, $d_f(x', x'')^2$ is equal to

$$\min_{\delta \in \mathbb{R}^{n_1}} \|\delta\|^2 \text{ subject to } H(x + \delta) = \mathbf{0}$$

where $x = \frac{x' - x''}{2}$. Then the lagrangian is

$$L(\delta, \lambda) = \delta^T \delta + \lambda^T H(x + \delta)$$

and the dual problem is

$$\max_{\lambda \in \mathbb{R}^m} x^T H^T \lambda + \min_\delta (\delta^T \delta + \lambda^T H \delta)$$
$$= \max_\lambda x^T H^T \lambda - \tfrac{1}{4} \lambda^T H H^T \lambda$$

where the minimizing $\delta$ is $-\frac{1}{2} H^T \lambda$. This is unconstrained, so it is easier to work with than the primal problem. Let $\lambda = c\mu$. For fixed $\mu$ with $\|\mu\| = 1$, the maximizing $c$ is $\frac{2x^T H^T \mu}{\mu^T H H^T \mu}$, so the dual can be reexpressed as

$$\max_{\mu: \|\mu\| = 1} \frac{(x^T H^T \mu)^2}{\mu^T H H^T \mu}$$

We have shown the matrix $HH^T$ appearing the denominator is close to $\|L\|_F^2 I$. This suggests that the optimizing the ratio is roughly the same as optimizing the numerator.

Pick $\mu^* = \frac{Hx}{\|Hx\|}$, $(x^T H^T \mu^*)^2 = \left( \frac{\|Hx\|^2}{\|Hx\|} \right)^2 = \|Hx\|^2 = \|ALx\|^2$.

$$\frac{\|Hx\|^2}{\max_{\mu: \|\mu\| = 1} \mu^T H H^T \mu} = \frac{(x^T H^T \mu^*)^2}{\mu^{*T} H H^T \mu^*} \leq \max_{\mu: \|\mu\| = 1} \frac{(x^T H^T \mu)^2}{\mu^T H H^T \mu} \leq \max_{\mu: \|\mu\| = 1} \frac{\|Hx\|^2 \|\mu\|^2}{\mu^T H H^T \mu} = \frac{\|Hx\|^2}{\min_{\mu: \|\mu\| = 1} \mu^T H H^T \mu}$$

Thus we need upper and lower bounds on $\|Hx\|^2$. Then $(ALx)_i \sim N(0, \|Lx\|^2)$ so $\|ALx\|^2$ is $\|Lx\|^2$ times a $\chi^2$ distribution with $m$ degrees of freedom. By Proposition 4, with $\|\sigma\|_2^2 = m\|Lx\|^2$, $\|\sigma\|_4^4 = m\|Lx\|^4$, $\|\sigma\|_\infty^2 = \|Lx\|^2$, and $r_4 = m$.

$$\Pr[\|ALx\|^2 \leq t] \leq -\frac{(t - m\|Lx\|^2)^2}{4m\|Lx\|^4}.$$

To get a bound of $\epsilon/2$, we take $t = m\|Lx\|^2 - \sqrt{4m\|Lx\|^4 \log(2/\epsilon)}$.

$$\frac{\|Hx\|^2}{\max_{\mu: \|\mu\| = 1} \mu^T H H^T \mu} \geq \frac{m\|Lx\|^2 - 2\|Lx\|^2 \sqrt{m \log(2/\epsilon)}}{\|L\|_F^2 + 4m\|L\|_4^2 \sqrt{\log(m^2/\epsilon)}} = \frac{m\|Lx\|^2}{\|L\|_F^2} \cdot \frac{1 - \frac{2}{\sqrt{m}} \sqrt{\log(2/\epsilon)}}{1 + \frac{4m\|L\|_4^2}{\|L\|_F^2} \sqrt{\log(m^2/\epsilon)}}$$

For $t \geq \|\sigma\|_2^2$, again using Proposition 4,

$$\log \Pr[\|ALx\|^2 \geq t] \leq \begin{cases} -\frac{(t - m\|Lx\|^2)^2}{16m\|Lx\|^4} & 0 \leq t - m\|Lx\|^2 \leq 2m\|Lx\|^2 \\ -\frac{t - 2m\|Lx\|^2}{4\|Lx\|^2} & 2m\|Lx\|^2 < t - m\|Lx\|^2 \end{cases}$$

To get a bound of $\epsilon/2$, we take

$$t = \begin{cases} m\|Lx\|^2 + 4\|Lx\|^2 \sqrt{m \log(2/\epsilon)} & \epsilon/2 \geq e^{-m/4} \\ 2m\|Lx\|^2 + 4\|Lx\|^2 \log(2/\epsilon) & \epsilon/2 > e^{-m/4} \end{cases}.$$

At the boundary between the cases, $t = 3m\|Lx\|^2$, which is three times the mean and too big of a threshold to interest us. Using only the first case, we get

$$\frac{\|Hx\|^2}{\min_{\mu: \|\mu\| = 1} \mu^T H H^T \mu} \leq \frac{m\|Lx\|^2 + 4\|Lx\|^2 \sqrt{m \log(2/\epsilon)}}{\|L\|_F^2 - 4m\|L\|_4^2 \sqrt{\log(m^2/\epsilon)}} = \frac{m\|Lx\|^2}{\|L\|_F^2} \cdot \frac{1 + \frac{4}{\sqrt{m}} \sqrt{\log(2/\epsilon)}}{1 - \frac{4m\|L\|_4^2}{\|L\|_F^2} \sqrt{\log(m^2/\epsilon)}}$$

$\square$

**Proposition 3.** *Let $H = AL$, where $L \in \mathbb{R}^{n_2 \times n_1}$ is a fixed linear feature extractor and the entries of $A \in \mathbb{R}^{m \times n_2}$ are i.i.d. standard Gaussian. Then for $\epsilon \geq m^2 e^{-r_4/4}$, for all $i, j \in [m]$,*

$$|(HH^T - \|L\|_F^2 I)_{ij}| \leq 4\|L\|_4^2 \sqrt{\log(2m^2/\epsilon)}.$$

*with probability $1 - \epsilon$.*

*Furthermore, for all $\mu \in \mathbb{R}^m$ with $\|\mu\| = 1$, $|\mu^T HH^T \mu - \|L\|_F^2| \leq 4m\|L\|_4^2 \sqrt{\log(2m^2/\epsilon)}$.*

*Proof.* Consider the singular value decomposition of $L$: $L = U\Sigma V^T$, where $U \in \mathbb{R}^{n_1 \times n_1}$. so $LL^T = U\Sigma^2 U^T$. Then $B = AU \in \mathbb{R}^{m \times n_1}$ also has i.i.d. standard Gaussian entries. We have $HH^T = ALL^T A^T = AU\Sigma^2 U^T A^T = B\Sigma^2 B^T$, so $(HH^T)_{ii} = \sum_{k=1}^{n_1} \sigma_k^2 B_{i,k}^2$. For $i \neq j$, $(HH^T)_{ij} = \sum_{k=1}^{n_1} \sigma_k^2 B_{i,k} B_{j,k}$. The first claim follows by applying Proposition 4 to $(HH^T)_{ii}$ and Proposition 5 case one to $(HH^T)_{ij}$. To achieve error probabilities of $\frac{\epsilon}{2m^2}$, we take $|t - E[(HH^T)_{ij}]| = 4\|L\|_4^2 \sqrt{\log(2m^2/\epsilon)}$.

For all $\mu \in \mathbb{R}^m$ with $\|\mu\| = 1$, using the triangle inequality we obtain

$$|\mu^T (HH^T - \|L\|_F^2 I)\mu| \leq \sum_{i,j} |\mu_i||\mu_j||(HH^T - \|L\|_F^2 I)_{ij}| \leq \langle|\mu|, \mathbf{1}\rangle^2 \cdot 4\|L\|_4^2 \sqrt{\log(2m^2/\epsilon)}$$

and $\langle|\mu|, \mathbf{1}\rangle^2 \leq \|\mu\|_2^2 \|\mathbf{1}\|_2^2 = m$. $\qquad\qquad\square$

**Proposition 4** (Chernoff bounds for linear combinations of $\chi^2$ random variables)**.** *Let $Z = \sum_{k=1}^n \sigma_k^2 B_k^2$, where $(B_k)$ are i.i.d. standard Gaussian. Then for $t \leq \|\sigma\|_2^2$*

$$\log \Pr[Z \leq t] \leq -\frac{(t - \|\sigma\|_2^2)^2}{4\|\sigma\|_4^4}$$

*and for $t \geq \|\sigma\|_2^2$*

$$\Pr[Z \geq t] \leq \begin{cases} -\frac{(t - \|\sigma\|_2^2)^2}{16\|\sigma\|_4^4} & 0 \leq t - \|\sigma\|_2^2 \leq 2r_4\|\sigma\|_\infty^2 \\ -\frac{(t - \|\sigma\|_2^2)}{4\|\sigma\|_\infty^2} + \frac{r_4}{4} & 2r_4\|\sigma\|_\infty^2 < t - \|\sigma\|_2^2 \end{cases}$$

*where $r_4 = \frac{\|\sigma\|_4^4}{\|\sigma\|_\infty^4}$.*

*Proof.* The cumulant generating function of $Z$ is

$$\log M(\theta) = \sum_{k=1}^n -\tfrac{1}{2} \log(1 - 2\sigma_i^2 \theta).$$

Observe that

$$E\left[\sum_{i=1}^n \sigma_i^2 B_{i,k}^2\right] = \sum_{i=1}^n \sigma_i^2 E[B_{i,k}^2] = \sum_{i=1}^n \sigma_i^2 = \|\sigma\|_2^2 = \|L\|_F^2$$

$$\text{Var}\left[\sum_{i=1}^n z_i^2 \sigma_i^2 B_{i,k}^2\right] = \sum_{i=1}^n \sigma_i^4 \text{Var}[B_{i,k}^2] = \sum_{i=1}^n 2\sigma_i^4 = 2\|\sigma\|_4^4 = 2\|L\|_4^4$$

**lower tail** For $z \leq 0$, $\log \frac{1}{1-z} \leq z + \frac{z^2}{2}$ (by comparing derivatives). We take $\theta^* = \frac{t - \|\sigma\|_2^2}{2\|\sigma\|_4^4}$ and for $t \leq \|\sigma\|_2^2$,

$$\log \Pr[(HH^T)_{ii} \leq t] \leq -\theta^* t + \frac{1}{2} \sum_{k=1}^{n_2} \log \frac{1}{1 - 2\sigma_i^2 \theta^*}$$

$$\leq -\theta^* t + \frac{1}{2} \sum_{k=1}^{n_2} (2\sigma_i^2 \theta^* + 2\sigma_i^4 \theta^{*2})$$

$$= -(t - \|\sigma\|_2^2)\theta^* + \theta^{*2}\|\sigma\|_4^4$$

$$= -\frac{(t - \|\sigma\|_2^2)^2}{4\|\sigma\|_4^4}$$

**upper tail** For $z < 1$, $\log \frac{1}{1-z} \leq \frac{z}{1-z}$ and for $z \leq \frac{1}{2}$, $\frac{z}{1-z} \leq z + 2z^2$. For all $\theta \geq 0$, $\Pr[(HH^T)_{ii} \geq t] \leq e^{-\theta t} M(\theta)$. Let

$$\theta^* = \min\left(\frac{t - \sum_{k=1}^{n_1} \sigma_k^2}{8 \sum_{k=1}^{n_1} \sigma_k^4}, \frac{1}{4 \max_k \sigma_k^2}\right) = \min\left(\frac{t - \|\sigma\|_2^2}{8\|\sigma\|_4^4}, \frac{1}{4\|\sigma\|_\infty^2}\right) = \begin{cases} \frac{t - \|\sigma\|_2^2}{8\|\sigma\|_4^4} & 0 \leq t - \|\sigma\|_2^2 \leq 2r_4\|\sigma\|_\infty^2 \\ \frac{1}{4\|\sigma\|_\infty^2} & 2r_4\|\sigma\|_\infty^2 < t - \|\sigma\|_2^2 \end{cases}.$$

This choice satisfies $2\sigma_i^2 \theta^* \leq \frac{1}{2}$ for all $i$, so we can apply our upper bound on $\log \frac{1}{1-z}$. Then for $t \geq \|\sigma\|_2^2$,

$$\log \Pr[(HH^T)_{ii} \geq t] \leq -\theta^* t + \frac{1}{2} \sum_{k=1}^{n_2} \log \frac{1}{1 - 2\sigma_i^2 \theta^*}$$

$$\leq -\theta^* t + \frac{1}{2} \sum_{k=1}^{n_2} (2\sigma_i^2 \theta^* + 8\sigma_i^4 \theta^{*2})$$

$$= -(t - \|\sigma\|_2^2)\theta^* + 4\theta^{*2}\|\sigma\|_4^4$$

$$= \begin{cases} -\frac{(t - \|\sigma\|_2^2)^2}{16\|\sigma\|_4^4} & 0 \leq t - \|\sigma\|_2^2 \leq 2r_4\|\sigma\|_\infty^2 \\ -\frac{(t - \|\sigma\|_2^2)}{4\|\sigma\|_\infty^2} + \frac{r_4}{4} & 2r_4\|\sigma\|_\infty^2 < t - \|\sigma\|_2^2 \end{cases}$$

$\square$

**Proposition 5** (Chernoff bounds for linear combinations of $\chi^2$ random variables). *Let $Z = \sum_{k=1}^n \sigma_k^2 B_k B_k'$, where $(B_k)$ and $(B_k')$ are i.i.d. standard Gaussian. Then*

$$\log \Pr[Z \geq t] \leq \begin{cases} -\frac{t^2}{16\|\sigma\|_4^4} & 0 \leq t \leq \sqrt{2}r_4\|\sigma\|_\infty^2 \\ -\frac{t}{\sqrt{2}\|\sigma\|_\infty^2} + \frac{r_4}{2} & \sqrt{2}r_4\|\sigma\|_\infty^2 < t \end{cases}$$

*Proof.* The cumulant generating function of $Z$ is

$$\log M(\theta) = \sum_{k=1}^n -\frac{1}{2} \log(1 - \sigma_i^4 \theta^2).$$

The off-diagonal entries have a symmetric distribution, so the upper and lower tail bounds are the same. For $z < 1$, $\log \frac{1}{1-z^2} \leq \frac{z^2}{1-z^2}$ and for $z^2 \leq \frac{1}{2}$, $\frac{z^2}{1-z^2} \leq 2z^2$. For $t \geq 0$, pick

$$\theta^* = \min\left(\frac{t}{2\|\sigma\|_4^4}, \frac{1}{\sqrt{2}\|\sigma\|_\infty^2}\right)$$

and

$$E\left[\sum_{i=1}^n \sigma_i^2 B_{i,k} B_{j,k}\right] = \sum_{i=1}^n \sigma_i^2 E[B_{i,k}] E[B_{j,k}] = 0$$

$$\text{Var}\left[\sum_{i=1}^n z_i^2 \sigma_i^2 B_{i,k} B_{j,k}\right] = \sum_{i=1}^n \sigma_i^4 E[B_{i,k}^2] E[B_{j,k}^2] = \sum_{i=1}^n \sigma_i^4 = \|\sigma\|_4^4 = \|L\|_4^4.$$

Then

$$\log \Pr[(HH^T)_{ij} \geq t] \leq -\theta^* t + \frac{1}{2} \sum_{k=1}^{n_2} \log \frac{1}{1 - \sigma_i^4 \theta^{*2}}$$

$$\leq -\theta^* t + \frac{1}{2} \sum_{k=1}^{n_2} 2\sigma_i^4 \theta^{*2}$$

$$= -\theta^* t + \theta^{*2} \|\sigma\|_4^4$$

$$= \begin{cases} -\frac{t^2}{16\|\sigma\|_4^4} & 0 \leq t \leq \sqrt{2} r_4 \|\sigma\|_\infty^2 \\ -\frac{t}{\sqrt{2}\|\sigma\|_\infty^2} + \frac{r_4}{2} & \sqrt{2} r_4 \|\sigma\|_\infty^2 < t \end{cases}$$

$\square$

