# OpenReview forum: "Evaluating the Adversarial Robustness of CNNs Layer by Layer"
_TMLR — Accepted by TMLR_

### Review · Reviewer_YcjA · 2025-11-09

**Summary Of Contributions:**

The paper presents a framework for analyzing the robustness of a feature extractor layer by layer. It proposes a novel, architecture-agnostic method for generating collusion adversarial pairs for a feature extractor. The analysis highlights the influence of various factors--such as training methods, activation functions and architecture--on layer-wise robustness of the feature extractor.

**Audience:**

Yes

**Audience Explanation:**

The paper presents a theoretical framework for analyzing the layer-wise robustness of a feature extractor. Researchers in the machine learning community working on adversarial robustness may find the observations in this work valuable.

**Claims And Evidence:**

No

**Claims Explanation:**

1. Presentation and organization of the paper: The current version of the manuscript is challenging to follow, particularly in Sections 2 and 3. The overall organization of the paper could be significantly improved to enhance clarity and flow.
2. Contributions: The present work builds upon Bhagoji et al. (2021, 2023). The authors  need to clearly highlight the advantages of the proposed method. Currently, the explanation of the proposed approach is entangled with the description of Bhagoji et al. (2021, 2023) in Section 2.
3. Validation experiment: The comparison between the proposed optimal collusion loss and the FAB loss (Section 5.1, Figures 5 and 6) might lead to incorrect interpretation. This is because the FAB attack generates the minimum perturbation required to change the class label of an input, whereas the proposed collusion algorithm generates adversarial pairs with similar feature representation-- and both the samples may not necessarily be misclassified.
4. Missing experimental details: The paper considers 100 samples from two classes in the training set of MNIST and CIFAR-10 datasets for evaluation. The details of selected classes and the justification for choosing a subset of the dataset are missing.  Additionally, the paper misses out to explain the impact of sample size on the robustness analysis.

**Requested Changes:**

1. [Critical] Please address the points discussed in the claims section, i.e., (a) Presentation and organization of the paper, (b) Contribution, (c) Validation experiment, and (d) Missing experimental details.
2. It would be beneficial to move the related works section after the introduction and include a section or subsection to describe the commonly used notation. Clearly highlight the advantages of the proposed method over existing methods.
3. Consider normally trained models for analysis. Also report the standard deviation for the proposed loss curves.

---

> ### Author Response · Authors · 2025-11-29
>
> **1. Presentation, Organization, and Contributions.**
>
> We have overhauled the paper structure to enhance flow and clearly delineate our novel contributions.
> - Restructuring: We moved the Related Work section to follow the Introduction and added a dedicated Notation subsection (Table 1) to define terms clearly before the methodology.
> - Clarifying Contributions: We rewrote related work to explicitly distinguish our work. While Bhagoji et al. provide the theoretical basis for feature robustness in discrete or low-dimensional settings, our work:
>   (i) Extends this theory to the high-dimensional setting of modern CNNs via Gaussian Random Projections.
>   (ii) Introduces a novel Gradient-based Collision Search (Algorithm 1) capable of finding collisions in deep, non-linear networks, overcoming the limitations of prior combinatorial approaches that scaled poorly with depth.
>
> The work Bhagoji et al. theory work stops at Section 3, our analysis of the high dimension with random projection starts from section 4. We have rewritten parts of Section 3 to make this more clear.
>
> ---
> **2. Comparison to FAB and interpretation of the loss.**
>
> We agree FAB and our method optimize different objectives. However, at the logit layer (Section 6.1), a collision implies $\tilde{x}$ and $\tilde{x}'$ share a predicted class. Since ground-truth labels differ ($y \neq y'$), at least one is misclassified. Thus, at the logits, our procedure produces valid adversarial examples. We revised the text to explicitly describe this relationship and frame FAB as a correlational sanity check rather than a direct benchmark. Our main conclusions regarding layer-wise robustness trends do not rely on this comparison.
>
> ----
>
> **3. Missing Experimental Details and Sample Size.**
>
> We thank the reviewer for pointing out this omission. We have updated the Experimental Setup to include the missing details and to analyze the effect of sample size:
>
> - Class selection.
>   To keep the analysis focused and reproducible, we fix a binary classification pair for each dataset. For MNIST and Cifar 10, we use digits 3 vs. 7. These class pairs are held fixed across all models and training schemes in the main experiments.
>
> - Subset size justification.
>   Our robustness metric is defined over all cross-class pairs, and for S samples per class this yields S^2 pairs. Since the Collision Search is run on every pair and on multiple random projections, the overall cost grows quadratically in S and quickly becomes prohibitive for large subsets. We therefore use S=100 samples per class (10,000 pairs) for the main experiments as a practical compromise between statistical stability and computational cost. We add this description in the main part.
>
>   It seems possible to develop more clever algorithms that use cheaply computed bounds to avoid the computation of all S^2 distances are possible, but we leave this as future work.
>
> - Impact of sample size and class choice.
>   To assess robustness of the conclusions, we perform a sensitivity analysis on the Medium CNN. First, we increase the subset size to S=200 per class (40,000 pairs). Second, we repeat the analysis with an alternative MNIST class pair (2 vs. 5). Third, we re-sample a different set of S=100 examples for the original class pair. As shown in Figure 22, the layer-wise robustness curves and the relative ordering of layers remain qualitatively unchanged across these variations, indicating that our conclusions are stable with respect to sample size and class choice. And we put this result to the appendix.
>
> ---
>
> **4. Standard Deviation and Normally Trained Models**
>
> - Standard Deviation: For two experiments, we have added error bars representing the standard deviation due to the random projection matrix for 7 layer medium CNN model. This was found by repeating with 15 random projection matrices. Figure 18 shows that the variance is low, confirming the metric's stability. And we put this result to the appendix.
>
> - Normally Trained Models: We analyzed normally trained (benign) models in the original submission. We have now highlighted these results (see Figure 13) to contrast them with PGD and TRADES training, showing that benign models lack significant feature space robustness even in early layers.

---

### Review · Reviewer_caFn · 2025-11-17

**Summary Of Contributions:**

The authors extend Bhagoji et al.’s feature-extractor robustness framework to wide CNNs by proposing a variation of the feature extractor distance for high-dimensional layers and implementing an approximate algorithm for ReLU CNNs. They introduce a novel collision-search algorithm and use random linear projections to evaluate each layer’s adversarial robustness contribution. In experiments on MNIST and CIFAR-10, the paper demonstrates that different training methods (e.g. PGD vs TRADES) lead to significant variation in layer-wise robustness, offering insight into how architectures and training decisions affect adversarial vulnerability.

**Audience:**

Yes

**Audience Explanation:**

The paper tackles adversarial robustness—an active and important research area—with new analysis tools and theory, which aligns well with TMLR’s focus. By providing a deeper layer-wise understanding of CNN robustness and a generic algorithm for feature-extractor analysis, the work offers broadly relevant insights for ML robustness and interpretability. Its combination of formal analysis and experimental findings is likely to engage readers interested in adversarial learning and model evaluation.

**Broader Impact Concerns:**

The paper does not discuss broader impact. In terms of ethics, there are no obvious negative implications beyond the general concerns of adversarial ML. It would be appropriate to note that layer-wise robustness analysis could help build safer models (positive impact) but also that understanding vulnerabilities might be misused by adversaries (negative). A brief Broader Impact statement could mention that improved robustness metrics advance security in applications (e.g. safety-critical systems) while cautioning about potential misuse for crafting attacks.

**Claims And Evidence:**

Yes

**Claims Explanation:**

The paper backs its claims with both theoretical analysis and empirical results. It presents a clear mathematical framework (with propositions and bounds) and provides algorithms for computing the robustness metric. Extensive experiments (Figures 5–15) on standard datasets verify the theoretical findings, showing concrete layer-by-layer robustness curves and validating the proposed distance metric. The evidence (e.g. observed robustness gaps between convolutional and fully-connected layers) is convincing and directly addresses the paper’s claims.

**Requested Changes:**

Clarify algorithmic details: The description of the collision-search algorithm and its implementation is somewhat terse. Including pseudocode or more detailed discussion (e.g. computational complexity, convergence criteria) would improve reproducibility.

Broaden evaluation: The experiments are limited to relatively small CNNs on MNIST/CIFAR-10. Evaluating on larger models or additional datasets (or providing a runtime/performance analysis) would strengthen the empirical claims.

Discuss limitations and comparisons: It would be helpful to compare the proposed metric with other robustness measures (or with and without random projection) and to discuss the impact of projection dimension. Also, commenting on the computational cost of the method and its scalability would be useful.

---

> ### Author Response · Authors · 2025-11-29
>
> **1. Algorithmic clarity and computational cost.**
>
> We agree that the original description of the collision-search algorithm and its implementation was too terse. In the revised version we have:
>
> * Clarified computational complexity: We now include an explicit complexity discussion in Section 4. For a given layer, the cost of computing our robustness metric over $N$ random projections and $T$ collision-search iterations scales as:
>     $$O\bigl(N \cdot T \cdot (\text{JacobianCost}(f_j, m) + m^2 n_0)\bigr)$$
>     where $\text{JacobianCost}(f_j, m)$ denotes the cost of the Jacobian computation via backpropagation for layer $j$, $m$ is the projection dimension, and $n_0$ is the input dimension. As we use small $m$ (e.g., $m=10$), the runtime is dominated by standard backprop through the feature extractor.
>
> * Reported runtime: We added a runtime table (Table 2 in the revision) that reports wall-clock time for MNIST and CIFAR-10 on Medium-CNN.
>
> * Specified convergence criteria: We now explicitly state the stopping rule for the collision search: the algorithm terminates when the change in the pair $(\tilde x, \tilde x')$ between iterations falls below a small Euclidean threshold (specifically $\|\Delta \tilde x\|_2 + \|\Delta \tilde x'\|_2 < 10^{-2}$ in our experiments).
>
> **2. Broadened evaluation and limitations.**
>
>  We agree that going beyond small CNNs on MNIST/CIFAR-10 is important for assessing generality.
>
> * Additional architectures: In the revised manuscript, we incorporate experiments on AlexNet and WideResNet-28-10 on CIFAR-10.
>
> * Limitations: We added an explicit limitations paragraph noting that our method relies on Jacobian computations, which become memory-intensive for very deep networks and high-resolution inputs. We therefore focus on MNIST and CIFAR-10 and on moderately deep CNNs in this work, and we highlight scaling to ImageNet-scale models as an important direction for future work, potentially requiring more memory-efficient Jacobian approximations.
>
> **3. Comparisons, projection dimension, and robustness measures.**
>
>  The current version of the paper already contains the requested comparisons. We compare our projected collision distance to the original full-dimensional feature-extractor distance of Bhagoji et al. (without projection), and show that in high-dimensional layers the latter yields almost no separation between layers, while our projected variant preserves clear robustness gaps. We also report an ablation over the projection dimension $m$ (e.g., $m=5,10,20,40$), finding that the layer-wise ordering is stable once $m$ is moderately large, with runtime growing roughly linearly in $m$. Finally, for the logit layer we relate our collision-based margin to standard adversarial margins (FAB), clarifying that although the objectives differ, they exhibit consistent qualitative trends. In the revision, we make these connections more explicit in the main text and point the reader to the relevant figures.

---

> > ### Comment · Reviewer_caFn · 2025-12-10
> >
> > Thank you for addressing my concerns. Please add the rebuttal clarifications in the final paper and also add a discussion about the broader impacts.

---

> > > ### Author Response · Authors · 2025-12-15
> > >
> > > Thank you for your helpful follow-up and for the positive assessment.
> > > We have incorporated the clarifications from our rebuttal directly into the revised manuscript (highlighted in blue), and we have added a discussion of the broader impacts on page 12.

---

### Review · Reviewer_SDCx · 2025-11-18

**Summary Of Contributions:**

The paper studies adversarial robustness at the level of individual layers in convolutional neural networks. Building on prior work that measures the robustness of low-dimensional feature extractors. This work extends the framework to high-dimensional CNN feature spaces via random projections and a conflict-graph-based optimal-loss computation. It proposes a practical algorithm that evaluates, for each layer, how robust its feature representation is under an $\ell_2$-norm bounded adversarial budget. They then conduct an empirical study across several CNN architectures and training schemes (standard, PGD, TRADES) on MNIST and CIFAR-10, relating layer-wise robustness profiles to training hyperparameters and to a freeze-k test that probes the usefulness of learned feature extractors.

**Additional Comments:**

The last page is blank. Could we remove that, please?

**Audience:**

Yes

**Audience Explanation:**

- Interesting and timely goal: understanding how robustness is distributed across layers rather than only at input or logits.

- Clear extension of prior theory to practical CNNs, with an implementable algorithm and reasonably well-organized experiments.

- The empirical layer-wise robustness curves and freeze-k experiments are intuitive and support the basic methodological claims.

**Broader Impact Concerns:**

**The paper does not include a Broader Impact Statement.**

Since adversarial robustness analysis is a dual-use topic, the work can be used both to strengthen defenses and to design more targeted attacks or model extraction strategies. A short Broader Impact section should be added that reflects on potential beneficial uses (for example, better detection and reinforcement of weak layers, more efficient robust training) as well as possible misuse, particularly in safety- and security-critical applications.

**Claims And Evidence:**

No

**Claims Explanation:**

Within the tested setting (CNNs on MNIST and CIFAR-10), the technical development and empirical results are coherent and the main qualitative observations about layer-wise robustness are reasonably supported.

However, the manuscript makes broader claims that are not convincingly justified by the current experiments or analysis. For example:

> The contributions section states: “We introduce a novel collision search algorithm applicable to any feature extractor.”

> Later, the related-work discussion adds: “our work introduces a novel collision search algorithm that can evaluate robustness for any feature extractor, regardless of the type of activation function used. This is achieved by leveraging gradient information, enabling the method to be widely applicable.”

However, only CNN feature extractors, that too just 2 unique CNN feature extractors have been experimented with, we do not know if the findings hold with deeper architectures, like WideResNet50, ConvNeXt-small/base, ResNet101, or with Vision Transformer-based extractors.

> In the methodology section, the Jacobian-subspace approach is further described as allowing the framework to “handle arbitrary activation functions and deep architectures efficiently and keep the result accuracy the same time.”

However, only one Activation function has been explored.

Additionally, all experiments are performed on convolutional networks of moderate depth, with low-resolution inputs, and there is no complexity analysis or empirical study on larger-scale architectures or datasets. The paper also focuses entirely on $\ell_2$ robustness, without discussion of $\ell_{\infty}$, which is also a standard in much of adversarial training practice. Because of this gap between the scope of claims and the presented evidence, the claims are not fully supported.

**Requested Changes:**

1. **Align generality and efficiency claims with presented evidence (Critical).**
   The manuscript should explicitly limit strong claims such as “applicable to any feature extractor”, “evaluate robustness for any feature extractor, regardless of the type of activation function used”, “enabling the method to be widely applicable”, and “handle arbitrary activation functions and deep architectures efficiently” to the regime that is actually studied (CNNs on MNIST and CIFAR-10), or provide additional evidence. Either temper these statements and mark broader applicability as future work, or add at least a few experiments on a more modern or deeper architectures and, ideally, a higher-resolution setting, with various activation functions.

2. **Clarify scalability and computational cost (Critical).**
   A more explicit discussion of computational and memory complexity is needed, for example, as a function of the number of layers, feature dimension, number of samples, and input resolution. Reporting representative runtimes or resource requirements for the main experiments would help assess whether the Jacobian-subspace collision search is realistically usable for larger models and datasets, and would substantiate claims of efficiency for deep architectures.

3. **Discuss the choice of $\ell_2$ norm and relation to $\ell_\infty$ (Critical).**
   The framework and experiments focus on $\ell_2$ robustness, while many existing adversarial training works also (or sometimes only) use $\ell_\infty$. The paper should justify the decision to work in $\ell_2$ (for example, mathematical tractability, empirical behaviour) and discuss whether the framework could be adapted to $\ell_\infty$. Even a conceptual discussion would help readers interpret the results relative to the also common $\ell_\infty$-robust models.

4. **Improve positioning with related work (Non-critical but important).**
   The related-work section would benefit from explicitly connecting to recent analyses of layer utilisation and robustness–fairness trade-offs, for example:
   - Gavrikov et al., “How Do Training Methods Influence the Utilization of Vision Models?” (NeurIPS 2024 Interpretable AI workshop) [1], which investigates how different training schemes, including adversarial training, change which layers are critical to performance.
   - Medi et al., “FAIR-TAT: Improving Model Fairness Using Targeted Adversarial Training” (WACV 2025) [2]. The direction of this work is quite different from the submission, since it focuses on targeted adversarial training for fairness, but its discussion of decision boundaries and how they are reshaped by adversarial training could provide useful context. A short comparison would clarify what new insight this collision-based metric adds beyond these perspectives.

5. **Address minor presentation issues (Non-critical).**
   Ensure that figures are referenced in order of appearance. At present, “Figure 4” appears in the text before “Figure 3”, and “Figure 2” is never referenced in the main text. Consider adding a brief limitations paragraph that clearly states the main scope restrictions (datasets, architectures, norm choice, and computational considerations).





REFERENCES

[1] Gavrikov, Paul, et al. “How Do Training Methods Influence the Utilization of Vision Models?” NeurIPS 2024 Workshop Interpretable AI: Past, Present and Future.

[2] Medi, Tejaswini, Steffen Jung, and Margret Keuper. “FAIR-TAT: Improving Model Fairness Using Targeted Adversarial Training.” 2025 IEEE/CVF Winter Conference on Applications of Computer Vision (WACV). IEEE, 2025.

---

> ### Author Response · Authors · 2025-11-28
>
> **1. Generality and activation functions**
> We agree that our original wording overstated the empirical scope of the study. While our collision-search algorithm relies only on first-order gradients---and is thus theoretically applicable to any differentiable feature extractor---we have revised the manuscript to explicitly restrict our empirical claims to CNN-based feature extractors on MNIST and CIFAR-10. We now mark broader applicability as a subject for future work.
>
> To address the concern regarding activation functions, we have added new experiments using GELU and Tanh activations on MNIST. As shown in Figure18 (included in the revised appendix), these experiments yield qualitatively similar robustness curves and convergence behaviors to the ReLU models. This supports our claim that the collision-search method is not strictly tied to piecewise-linear activations like ReLU.
>
> ---
>
> **2. Scalability and computational cost**
>
> We have added a more explicit complexity discussion to the main paper. Our algorithm alternates between:
>
> 1. Computing local Jacobians to linearize the collision constraint at a given layer.
> 2. Projecting onto the resulting affine subspace.
>
> - **Time complexity.** The total cost to find a collision pair scales as
>   $$O\Big(N \cdot T \cdot (\text{JacobianCost}(f_j, m) + m^2 n_0)\Big),$$
>   where \(N\) is the number of random projections, T is the number of iterations, n_0 is the input dimension, and m is the projection dimension (we use m = 10 by default). In practice, the dominant cost is the backpropagation required to compute the Jacobians.
>
> - **Memory complexity.** Memory usage is dominated by storing activations for backpropagation and scales as
>   $$O(\text{batch size} \times \text{network depth} \times \text{feature resolution})$$.
>   For our 7-layer CNN on a 40GB GPU, we can process batches of 100 pairs; for deeper ResNet-style models we reduce the batch size to around 15 pairs to stay within memory limits.
>
> We also report representative wall-clock runtimes in the revised paper. We declare convergence we use in the experiment is that when the L2 norm between the two images in the adversarial pair falls below 0.01.
>
> ---
> 3. **Algorithmic feasibility and adaptation to L-infinity.**
> - Distance structure of the data: We focus on L2-constrained adversaries because the L2 distance structure of the data is more interesting than the L-infinity structure. Almost all pairs of examples are very well-separated in the L-infinity distance, so there is some classifier that can achieve very high accuracy even when the adversary has a large budget. Thus the lower bounds on optimal losses tend to be lower in the L-infinity case. There is typically a larger gap between the lower bounds and observed robustness accuracy for L-infinity relative to L2.
>
> - Theoretical Compatibility: Our framework relies on random projections to estimate properties of the feature space. However, the rotational invariance of these projections is not connected to the rotational invariance of L2 balls in the input space. The random projection approach still makes sense for measuring L-infinity robustness. However, the proof of our distance concentration result for linear feature extractors would need significant modification to produce an L-infinity version of the statement. We could probably be do it if required.
>
> - Algorithmic Feasibility: Our collision search (Algorithm 1) could be adapted to L-infinity constraints. We would need to replace the orthogonal projection to an affine subspace step with finding the closest point in L-infinity distance, but this can be done by solving a simple linear program. The subsequent step of calculating the optimal lower-bound loss (via the conflict graph) is completely independent of the adversarial constraint type.
>
> We have added a "Limitations" section explicitly stating that our current theoretical guarantees are optimized for the L2 setting.
>
> ---
>
> **4. Positioning with related work**
>
> We appreciate the reviewer highlighting these relevant studies, and we have updated our Related Work section to explicitly position our contributions against them. Specifically, while Gavrikov et al.\ (NeurIPS 2024 Workshop) investigate layer utilization and criticality from a functional pruning perspective, our work offers a complementary geometric perspective by analyzing feature space collisions. Similarly, while Medi et al.\ (WACV 2025) explore how targeted adversarial training reshapes decision boundaries for fairness, our framework focuses on untargeted robustness lower bounds.
>
> ---
>
> **5. Minor issues**
>
> We have corrected the figure numbering in the paper and removed the trailing blank page.

---

### Decision · Action_Editor_fyzL · 2025-12-20

**Recommendation:** Accept with minor revision

**Audience:**

Yes

**Audience Explanation:**

This paper will be of use to the adversarial robustness community.

**Claims And Evidence:**

Yes

**Claims Explanation:**

The paper presents a well-motivated and technically rigorous framework for layer-wise adversarial robustness analysis, providing both theoretical insights and empirical validation on standard benchmarks. While some reviewers highlighted limitations in experimental breadth and scope, the methodology is sound, and the core observations are clearly supported within the studied settings, making the work relevant to the robustness community. The revised manuscript improves clarity, appropriately tempers broader claims, and better situates the contribution within prior literature. The remaining concerns regarding scalability, generality, and presentation can be addressed through clearer discussion and contextualization rather than substantial technical revisions. Overall, the work is timely, informative, and suitable for acceptance subject to minor revision.